# Urinary proteome profiling for stratifying patients with familial Parkinson's disease

Sebastian Virreira Winter[1,†,‡] (iD), Ozge Karayel[1,†] (iD), Maximilian T Strauss[1,†], Shalini Padmanabhan[2], Matthew Surface[3], Kalpana Merchant[4], Roy N Alcalay[3] & Matthias Mann[1,5,*] (iD)

## Abstract

The prevalence of Parkinson's disease (PD) is increasing but the development of novel treatment strategies and therapeutics altering the course of the disease would benefit from specific, sensitive, and non-invasive biomarkers to detect PD early. Here, we describe a scalable and sensitive mass spectrometry (MS)-based proteomic workflow for urinary proteome profiling. Our workflow enabled the reproducible quantification of more than 2,000 proteins in more than 200 urine samples using minimal volumes from two independent patient cohorts. The urinary proteome was significantly different between PD patients and healthy controls, as well as between *LRRK2* G2019S carriers and non-carriers in both cohorts. Interestingly, our data revealed lysosomal dysregulation in individuals with the *LRRK2* G2019S mutation. When combined with machine learning, the urinary proteome data alone were sufficient to classify mutation status and disease manifestation in mutation carriers remarkably well, identifying VGF, ENPEP, and other PD-associated proteins as the most discriminating features. Taken together, our results validate urinary proteomics as a valuable strategy for biomarker discovery and patient stratification in PD.

**Keywords** biomarker; DIA; mass spectrometry; Parkinson's disease; urinary proteome

**Subject Categories** Biomarkers; Neuroscience; Proteomics

## Introduction

With a population prevalence of 0.2%, Parkinson's disease (PD) is the second most common neurodegenerative disorder after Alzheimer's disease (De Lau & Breteler, 2006). It is characterized by the progressive loss of dopaminergic neurons and accumulation of α-synuclein-containing protein aggregates called Lewy bodies in the cytoplasm of the remaining neurons. As a result of dopaminergic neuron loss, PD manifests with motor signs and symptoms including bradykinesia, tremor, and rigidity, and these characteristics are used for diagnosing the disease (De Lau & Breteler, 2006; Reeve *et al*, 2014; Tysnes & Storstein, 2017).

PD is a genetically complex disorder. Most patients do not carry a single pathogenic variant linked to PD, but a subset of about 10% of patients carry an identifiable pathogenic variant in genes such as *SNCA*, *PRKN*, *LRRK2*, or *GBA*. For these individuals, the risk of developing the disease increases to 2–5% (Reeve *et al*, 2014). Among these genes, *LRRK2* is relatively common and causes PD in an autosomal dominant with incomplete penetrance fashion. How *LRRK2* mutations cause PD is unknown, however, several studies have indicated that disease-linked *LRRK2* mutations elevate its kinase activity and thereby contribute to PD pathogenesis (West *et al*, 2005; West, 2015). We have previously identified multiple Rab GTPases as endogenous targets of mutant LRRK2 (Steger *et al*, 2016; Steger *et al*, 2017; Karayel *et al*, 2020). Furthermore, inhibitors of this kinase have emerged as promising therapeutics for PD and clinical trials have already passed phase 1(Tolosa *et al*, 2020). Although idiopathic forms of PD presumably represent a heterogeneous collection of pathogenic mechanisms, *LRRK2*-associated PD and idiopathic PD (iPD) show a similar phenotype in terms of disease symptoms and response to levodopa. The interest in LRRK2 as a therapeutic target is also fueled by the association between common variants in *LRRK2* and sporadic PD (Nalls *et al*, 2019) and the observation that LRRK2 activity is increased in autopsied brain tissue from iPD patients without a known pathogenic mutation (Di Maio *et al*, 2018). Thus, it will be important to conduct studies on ante-mortem biospecimens to gain insights into *LRRK2* mutation-induced changes and thereby identify iPD patients who could benefit from LRRK2-targeted therapies.

Current treatments, including levodopa—the most effective PD medication, mainly alleviate the motor symptoms rather than slow

1   Department of Proteomics and Signal Transduction, Max Planck Institute of Biochemistry, Martinsried, Germany
2   The Michael J. Fox Foundation for Parkinson's Research, New York, NY, USA
3   Department of Neurology, Columbia University, New York, NY, USA
4   Northwestern University Feinberg School of Medicine, Chicago, IL, USA
5   Novo Nordisk Foundation Center for Protein Research, Faculty of Health Sciences, University of Copenhagen, Copenhagen, Denmark
    *Corresponding author. Tel: +49 89 8578 2557; E-mail: mmann@biochem.mpg.de
    †These authors contributed equally to this work
    ‡Present address: OmicEra Diagnostics GmbH, Planegg, Germany

disease progression or reverse the pathology. Given the growing number of PD patients worldwide, and escalating economic and societal implications, there is an urgent need for disease-modifying therapeutics. The development of new therapeutic strategies requires better insights into the pathophysiologic mechanisms of PD as well as biomarkers to detect the earliest stages of PD before severe motor impairment is evident and irreversible brain damage has already occurred. Although cerebrospinal fluid (CSF) has been frequently used for biomarker studies of brain disorders, recent studies indicate that urine offers another promising clinically viable matrix for PD since it can be frequently and non-invasively collected in large volumes (Alcalay *et al*, 2020). Importantly, urine contains not only kidney and urinary tract proteins but also filtered plasma proteins originating from distal organs, including the brain (Decramer *et al*, 2008; An & Gao, 2015). Therefore, urine protein analysis may provide diagnostic and prognostic opportunities for both urogenital and non-renal diseases (Kaiser *et al*, 2004; Mischak *et al*, 2004; Nguyen *et al*, 2005; Tantipaiboonwong *et al*, 2005; Adachi *et al*, 2006; Ward *et al*, 2008; Zimmerli *et al*, 2008; Kentsis *et al*, 2009; Nagaraj & Mann, 2011; Kentsis *et al*, 2013; Metzger *et al*, 2013; Duangkumpha *et al*, 2019; Ferrari *et al*, 2019). Recent technical advances in fast and high-throughput sample preparation methods in conjunction with improvements in high-accuracy mass spectrometry (MS)-based proteomics have enabled characterization of the urinary proteome (Berger *et al*, 2015; Batth *et al*, 2019; Ding *et al*, 2020). However, to what extent neurodegenerative disorders including PD affect the urinary proteome remains unknown.

Our group has recently employed state-of-the-art MS-based proteomics to obtain proteome profiles of the two body fluids plasma and CSF in multiple disease conditions (Geyer *et al*, 2016a; Geyer *et al*, 2016b; Albrechtsen *et al*, 2018; Geyer *et al*, 2019; Niu *et al*, 2019). Here, we extend this technology to urinary proteome profiling and provide first evidence that this approach can be used for PD biomarker discovery. More specifically, we focused our analysis on two large patient cohorts, both including healthy control subject, non-manifesting carriers of the frequently found *LRRK2* G2019S mutation, manifesting patients with the same mutational signature, and PD patients without the *LRRK2* mutation. The composition of the cohorts, quality of the data, and the depth of proteome coverage allowed us to identify pathogenic *LRRK2*-regulated lysosomal protein signatures that could serve as biomarkers to stratify subjects with pathogenic *LRRK2*. Taken together, our study offers evidence that quantitative MS-based proteomics represents a clinically useful strategy for non-invasive monitoring of disease progression and treatment response as well as patient stratification in PD.

# Results

### Overview of PD cohorts for urinary proteomics

Here, we employ a recently described "rectangular" biomarker discovery strategy in which as many proteins as possible are measured using shotgun MS-based proteomics for all the individuals in both discovery and validation cohorts (Geyer *et al*, 2016b; Geyer *et al*, 2017). To map proteome changes between individuals with different mutation status and manifestation of disease, we analyzed 235 urine samples from two independent cross-sectional cohorts

each comprised of four subject groups: (i) healthy controls (HC, $LRRK2^-/PD^-$); (ii) non-manifesting carriers (NMC) harboring the *LRRK2* G2019S mutation ($LRRK2^+/PD^-$); (iii) idiopathic PD patients (iPD, $LRRK2^-/PD^+$); and (iv) manifesting PD patients with *LRRK2* G2019S (LRRK2 PD, $LRRK2^+/PD^+$; Fig 1A and Table 1).

The first cohort was recruited at Columbia University Irving Medical Center (hereinafter referred to as "Columbia cohort" and color-coded with orange; Alcalay *et al*, 2020; Melachroinou *et al*, 2020). Participants in the Columbia cohort included 35 HC, 16 NMC, 40 iPD, 28 LRRK2 PD individuals, and one PD patient with an unknown *LRRK2* status. The cohort included 52 female sex and 68 male sex individuals (Fig 1A and Table 1). The *GBA* (gene that encodes for lysosomal acid glucosylceramidase (GCase)) mutation status was also available for all individuals, with 22 of them harboring a *GBA* variant and 98 the wild-type allele. $PD^+$ and $PD^-$ subjects were frequency-matched by age with means of $67.0 \pm 9.3$ and $64.1 \pm 12.0$ ($\pm$ SD) years, respectively (Appendix Fig S1A). Their motor skills were assessed using the Unified Parkinson's Disease Rating Scale part III (UPDRS-III) and cognitive functioning with the Montreal Cognitive Assessment (MoCA) test (Appendix Fig S1B and C). Genotyping for *LRRK2* G2019S mutation was conducted as previously described (Alcalay *et al*, 2015).

To confirm findings from the Columbia cohort, we additionally analyzed a subset of biobanked urine samples from the Michael J. Fox Foundation for Parkinson's Research (MJFF)-funded *LRRK2* Cohort Consortium (LCC) (hereinafter referred to as "LCC cohort" and color-coded with blue). We determined urine proteomes for 26 HC, 37 NMC, 29 iPD, and 23 LRRK2 PD individuals (53 female and 62 male) (Fig 1A and Table 1). In the LCC cohort, individuals in the non-diseased group were somewhat younger ($53.8 \pm 13.9$ years) than PD patients ($67 \pm 7.6$ years) (means $\pm$ SD, Appendix Fig S1D). In addition, LCC sample collection protocols were less stringent than in the Columbia cohort and UPDRS-III and MoCA scores were not available, indicating that the Columbia cohort is more powerful for our analyses. Both studies were approved by local institutional review boards, and each participant signed an informed consent (see Dataset EV1 for a detailed overview).

### Proteomic characterization of urine samples

For the proteomic profiling of individual urine samples, we developed a high-throughput proteomics workflow building on the PVDF-based sample processing method MStern blotting by the Steen group (Berger *et al*, 2015) combined with data-independent acquisition (DIA) LC-MS/MS (Gillet *et al*, 2012; Ludwig *et al*, 2018; Fig 1A). To maximize proteome depth, we generated two cohort-specific hybrid spectral libraries by merging three sub-libraries: (i) a library constructed by data-dependent acquisition (DDA) consisting of 24 fractions of pooled neat urine samples; (ii) a DDA library consisting of 8 fractions of extracellular vesicles isolated from pooled neat urine samples; and (iii) a directDIA library generated from the DIA analysis of all analyzed samples (see Materials and Methods). In these hybrid libraries, we identified a total of 4,564 and 5,725 protein groups for the Columbia and LCC cohorts, respectively (Appendix Fig S1E). Applying this robust workflow, we quantified on average 2,026 (Columbia) and 2,162 (LCC) protein groups per neat urine sample, in single runs of 45 min and using less than 100 μl of starting material (Dataset EV2). Three outlier samples

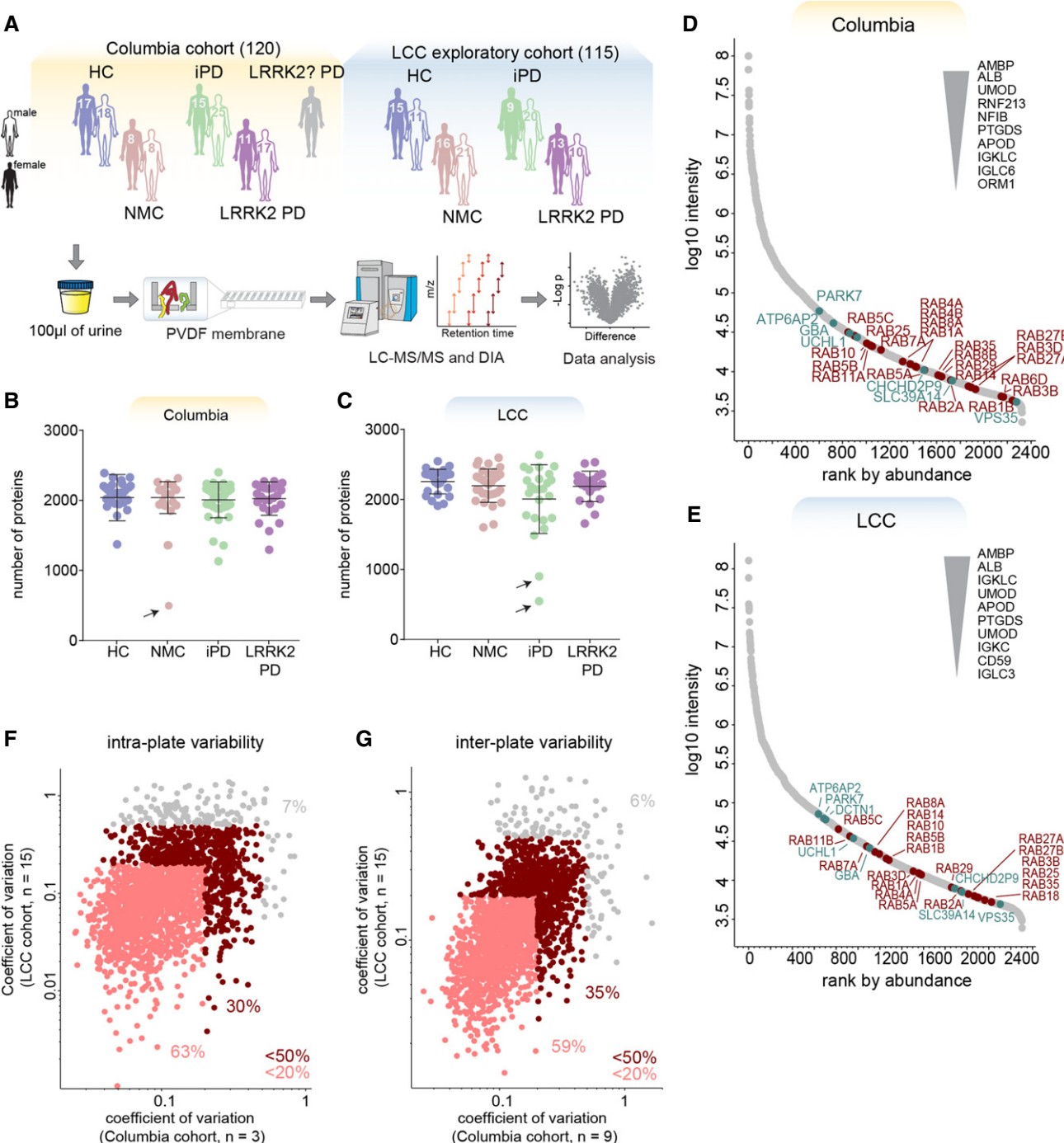

**Figure 1. MS-based proteomic analysis of two independent urinary PD cohorts has high depth and precision.**

A    Overview of the two cohorts and the proteomic workflow. Urine samples comprised of four subject groups ((HC, iPD, NMC and LRRK2 PD) were prepared using MStern blotting and analyzed by LC-MS/MS using data-independent acquisition (DIA). The sex and total number of subjects per cohort group is shown.

B, C    Number of proteins identified and quantified with a 1% false-discovery rate (FDR) in each sample in the Columbia (B) and LCC (C) cohorts. Bars indicate mean and standard deviation. Arrows point at one subject from the Columbia and two subjects from the LCC cohort that were excluded from further analysis due to low proteome depth.

D, E    Proteins identified in the Columbia (D) and LCC (E) cohort were ranked according to their MS signals, which covered more than five orders of magnitude. The top ten most abundant, Parkinson-related proteins (green) and Rab GTPases (red) are labeled.

F, G    Quantification precision assessed by calculating the intra-plate (F) and inter-plate (G) coefficients of variation (CVs) for the Columbia and LCC cohorts. Proteins with a CV below 20% and 50% in both cohorts are highlighted in light and dark red, respectively and the fractions of proteins above and below these CV thresholds are shown. A total of 2,051 proteins were consistently quantified in both cohorts.

**Table 1. Demographics of all participants**

| Columbia cohort | HC (*LRRK2*−/PD−) (n = 35) | NMC (*LRRK2*+/PD−) (n = 16) | iPD (*LRRK2*−/PD+) (n = 40) | LLRK2 PD (*LRRK2*+/PD+) (n = 28) |
|---|---|---|---|---|
| Age at collection, mean (SD) | 67.5 (10.3) | 56.8 (12.7) | 64.9 (9.2) | 70.7 (8.7) |
| Age at onset, mean (SD) | n/a | n/a | 57.8 (11) | 57.9 (11.3) |
| Sex (female/male) | 17/18 | 8/8 | 15/25 | 11/17 |
| *GBA* (mut/WT) | 7/28 | 1/15 | 11/29 | 3/25 |
| MoCA, mean (SD) | 27.5 (2) | 28.7 (1.1) | 26.9 (1.6) | 26.3 (4.5) |
| UPDRS-III, mean (SD) | 1.1 (1.5) | 0.8 (1.1) | 17.7 (10) | 20.6 (8) |
| **LCC cohort** | **HC (*LRRK2*−/PD-)** (n = 26) | **NMC (*LRRK2*+/PD−)** (n = 37) | **iPD (*LRRK2*−/PD+)** (n = 29) | **LLRK2 PD (*LRRK2*+/PD+)** (n = 23) |
| Age at collection, mean (SD) | 56.1 (16) | 52.1 (12.1) | 65 (9.1) | 68.4 (5) |
| Age at onset, mean (SD) | n/a | n/a | 58.4 (9.1) | 57.7 (7.6) |
| Sex (female/male) | 15/11 | 16/21 | 9/20 | 13/10 |

were excluded from further analysis due to low proteome depth (Fig 1B and C, Dataset EV1). The quantified protein intensities spanned five orders of magnitude in both cohorts and the top ten most abundant proteins contributed about half to the total urinary proteome signal (Fig 1D and E). As observed before (Adachi *et al*, 2006), the molecular weight distribution spanned a wide range with many proteins exceeding 100 kDa. More than 2,000 proteins were in common between the two cohorts. To the best of our knowledge, this study presents the deepest urinary proteome coverage for single-run analysis to date, a promising basis for the discovery of biomarkers.

Data from repeated measurements of individual samples revealed a high reproducibility with more than 90% of proteins having an intra- and inter-plate coefficient of variation (CV) below 50% in both studies and about 60% of proteins with a CV below 20% (Fig 1 F and G, Appendix Fig S2). The intra- and inter-plate variability within each cohort was even lower (Appendix Fig S2C and I), while the inter-individual variability was much larger with no protein having a CV below 20% (Appendix Fig S2F and L). Thus, our proteomic quantification precision greatly exceeds the biological variability that we seek to measure.

**Quality assessment of urine samples**

Pre-analytical variation caused by inconsistent sample processing and contaminations during sample collection can have a strong impact on the results and may cause the reporting of incorrect biomarkers (Geyer *et al*, 2019). To ensure that the observed proteome changes are not caused by artifacts related to sample handling and processing, we assessed each sample for potential quality issues. To this end, we used a previously reported quality marker panel to determine the degree of contamination with erythrocytes (Geyer *et al*, 2019; Fig 2A and B and Dataset EV3). Insufficient removal of cells and cellular debris from urine leads to an increased detection of intracellular proteins with a high sample-to-sample variability compared to regularly secreted urinary proteins (Guo *et al*, 2015). We therefore generated a second urine-specific quality marker panel to assess the degree of contamination with cells and cellular debris that could originate from aged,

inflamed, or damaged tissue of the kidneys, bladder, or the urinary tract (see Materials and Methods). Although urine samples from both cohorts were cleared by centrifugation following collection to avoid this systematic bias, our procedure flagged four samples from the Columbia cohort for potential contamination with cellular components (Fig 2A and B). Taken together, 6 samples from the Columbia cohort and 4 samples from the LCC cohort showed increased intensities of contamination markers and were thus excluded from further analyses. In addition, we further excluded one sample from the Columbia cohort, as it clustered far away from all other samples in a principal component analysis (PCA), likely indicating pre-analytical variation.

Next, we generated a global correlation map of the urinary proteome to identify clusters of functional co-regulation as previously reported for plasma proteome profiling (Albrechtsen *et al*, 2018). The global correlation map contains pairwise relations of all urinary proteins across 112 samples from the Columbia cohort. Unsupervised hierarchical clustering of the pairwise Pearson correlation coefficients revealed four main and several small clusters of co-regulated proteins (Fig 2C). The largest of these clusters was chiefly enriched for proteins with the Gene ontology (GO)-term "extracellular exosome" as well as other significant terms (Dataset EV4). A more detailed investigation of exosomes, their synthesis and secretion may thus be interesting in future studies. We also identified a cluster of highly correlated proteins that was enriched for the GO terms "immunoglobulin" and "B-cell receptor", suggesting that these proteins originate from immune cells. The two further main clusters were enriched for proteins originating from sex-specific tissues such as the prostate and vagina (Fig 2C; Uhlén *et al*, 2015). This shows that sex-dependent anatomical differences strongly affect the urinary proteome and thus should be considered as confounding factors. Indeed, a principal component analysis indicated sex as the strongest contributor to the inter-individual variance of the urinary proteome (Fig 2D and E).

**Detection of PD-related proteome alterations in urine**

Although PD primarily manifests in the central nervous system and is characterized by motor impairments, it is known to affect

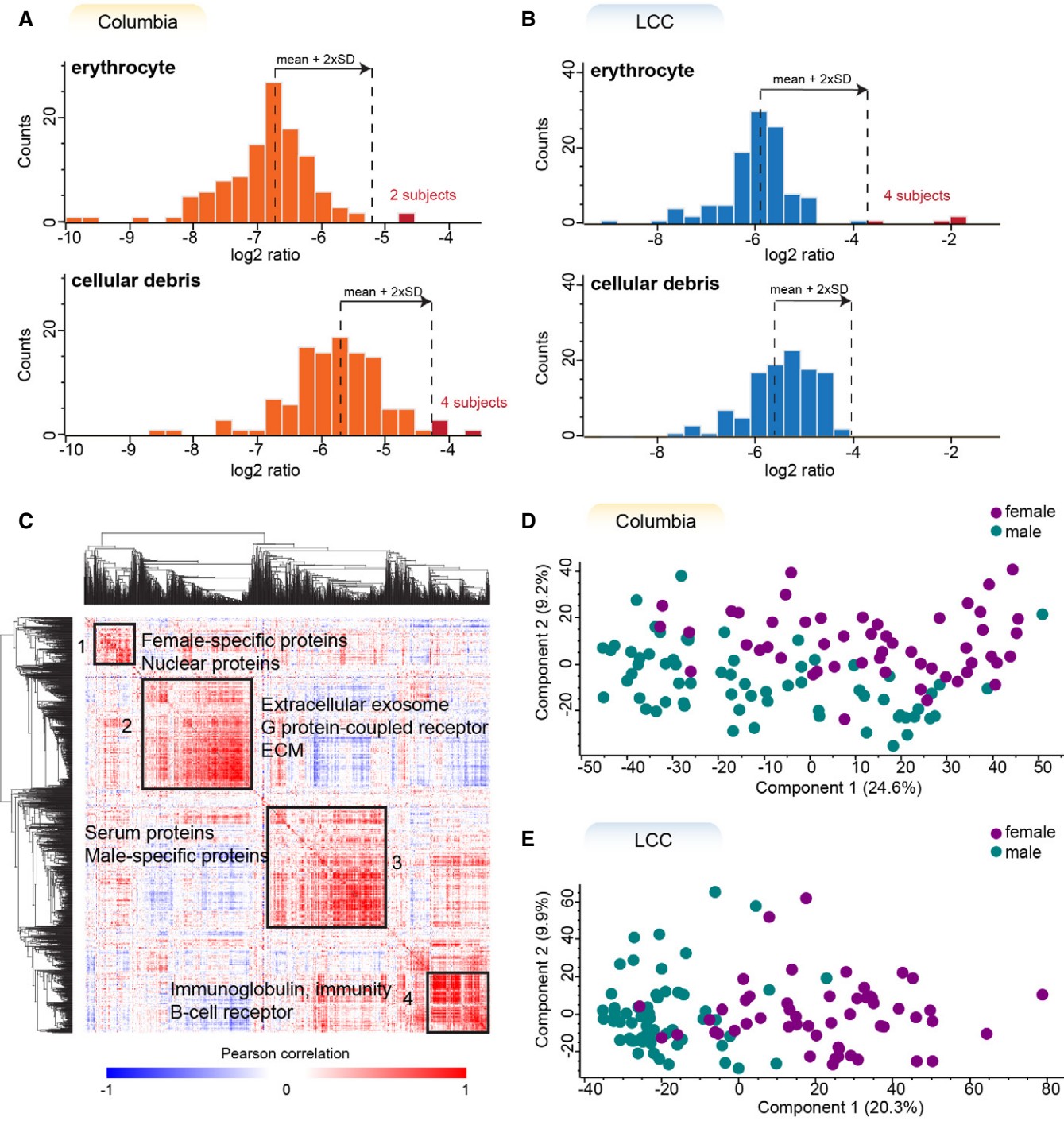

**Figure 2. The large majority of urine samples has high quality and shows sex-specific protein expression.**

A, B   Histograms of log$_2$ transformed ratios of the summed intensity of the proteins in the respective quality marker panel and the summed intensity of all proteins in Columbia (A) and LCC (B) cohorts. A sample was flagged for potential contamination and removed from further analysis if the ratio differed more than two standard deviations from the mean of all samples within the cohort. The proteins in each quality marker panel are listed in Dataset EV3.

C       Global correlation map of proteins generated by clustering the Pearson correlation coefficients of all pairwise protein comparisons for the Columbia cohort.

D, E   Principal component analysis (PCA) of all subjects based on their urinary proteome profiles. Female subjects are shown in purple and males in green.

and potentially initiate in peripheral tissues and is associated with non-motor symptoms (Poewe, 2008; Jain, 2011). Thus, we asked if the disease also causes changes of the urinary proteome, which reflects proteins from both central and peripheral organs. To identify PD-associated changes in the urine proteome, we first determined which proteins are differentially present in the urine of PD

patients compared to the controls, irrespective of their *LRRK2* status. To control for confounders, we performed an analysis of covariance (ANCOVA) considering sex, age at sample collection, *LRRK2* status, and *GBA* status (only available for the Columbia cohort) as confounding factors. Applying a 5% false-discovery rate (FDR) cutoff, we identified 361 proteins that displayed significantly different levels in PD patients when compared to controls (298 in Columbia cohort and 73 in LCC cohort) (Dataset EV5). The relatively large number of regulated proteins is in agreement with previous reports that the majority of PD patients suffers from urinary tract dysfunction (Yeo *et al*, 2012). The smaller number of significantly different proteins in the LCC cohort as well as the relatively small overlap between the cohorts could be explained by a less stringent sample collection protocol and worse age-matching in the LCC cohort. The $\log_2$ fold-changes between PD patients and non-diseased individuals show a good correlation between the two cohorts (Pearson $r = 0.65$) (Fig 3A), reflecting both reproducibility of the applied proteomic workflow and pathobiological consistency. The mean fold-changes of the 330 PD-associated proteins that were quantified in both cohorts were larger for the Columbia cohort (Columbia: 1.43 (up) and 0.49 (down) vs. LCC: 1.27 (up) and 0.75 (down)). Furthermore, 90% of the PD-associated proteins were detected with at least two peptides and quantified with CVs below 50% (Appendix Fig S3A).

Protein misfolding is known to be involved in many neurodegenerative conditions including PD (Cook *et al*, 2012). Interestingly, some of the proteins exhibiting the largest differential levels between the urine of controls vs. PD patients include proteins assisting other proteins in folding, such as peptidyl-prolyl cis-trans isomerase B (PPIB) and T-complex protein 1 subunit gamma (CCT3) (Fig 3A). We also identified two of the eight human canonical ribonucleases (RNASE1 and RNASE2) to be PD-associated in both cohorts (Fig 3A). The levels of the four apolipoproteins APOA1, APOA2, APOA4, and APOC1 were also elevated in PD patients (Fig 3A). While they show a similar trend in both cohorts, they reached statistical significance only in the Columbia cohort, corroborating that this cohort has greater power to detect PD-associated changes.

Next, we analyzed if any GO terms assigned to the 361 PD-associated proteins were significantly enriched compared to the urinary proteome (Fig 3B). This analysis examines if PD affects individual cellular compartments and particular biological signaling networks in urine. The term "bone development" was significantly enriched, in line with previous findings that PD patients are at increased risk for osteoporosis and osteopenia (Torsney *et al*, 2014). In summary, we observed disease-associated protein signatures with a high correlation between the two independent cohorts and identified promising candidates that could serve as biomarkers for PD and provide mechanistic insights into disease pathogenesis.

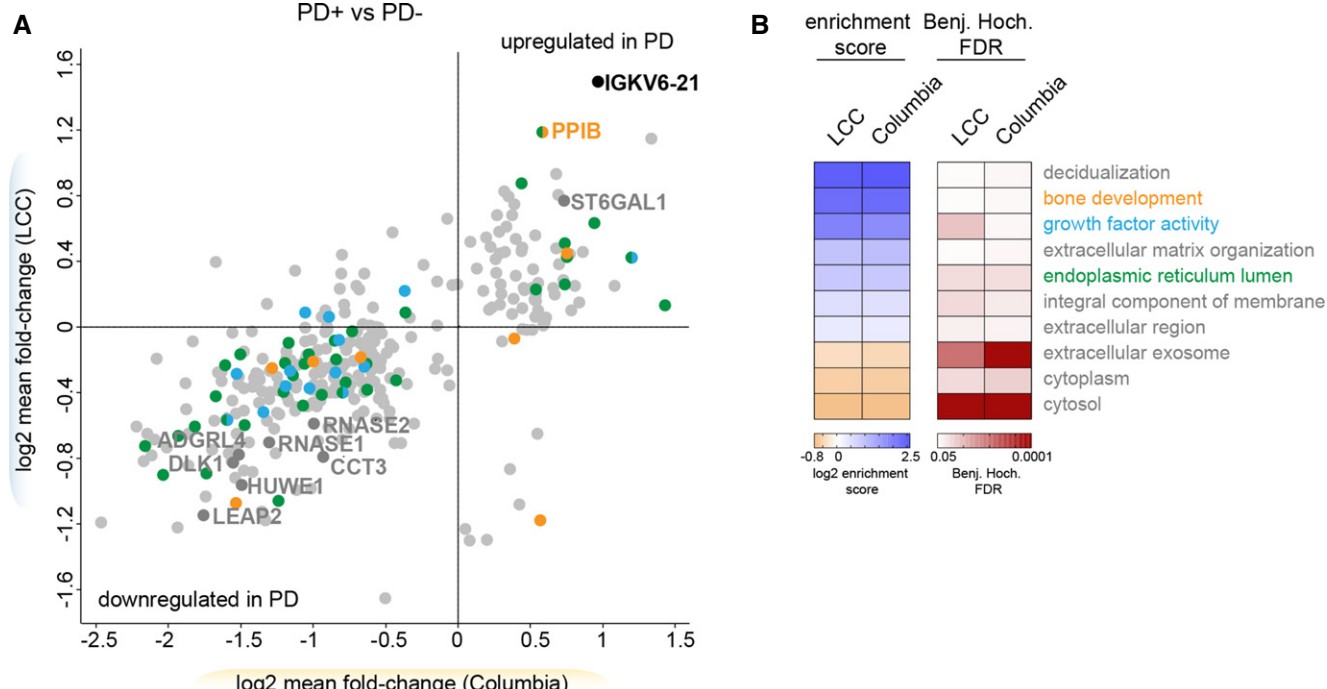

**Figure 3. PD affects the urinary proteome.**

A  Correlations of mean fold-changes of the proteins changing PD-dependently in the Columbia and LCC cohorts. Only proteins quantified in both cohorts are shown ($n = 330$). The colors match to the GO terms shown in (B). Proteins overlapping between the two cohorts are labeled with their name.

B  Fisher exact test to identify significantly enriched GO terms in the PD-associated proteins in urine. Importantly, the enrichment score of the Fisher exact test does not indicate if the proteins were up- or downregulated in PD patients but rather that the regulated proteins—independent of the directionality—compared to the total urinary proteome are associated with the enriched term. All GO terms that were significant in both cohorts are displayed (FDR < 5%).

## Pathogenic *LRRK2*-dependent changes are linked to lysosomes and glycosphingolipid metabolism

Encouraged by the observation of disease-dependent proteome changes in urine, we next asked if the urinary proteome is altered by the presence of the *LRRK2* G2019S mutation. We again applied an ANCOVA analysis with sex, age at sample collection, PD status, and *GBA* status (only available for the Columbia cohort) as confounding factors and compared the proteomes between G2019S and wild-type allele carriers. Applying an FDR of 5%, the mutation altered the abundance of 237 proteins (Columbia: 166, LCC: 104) (Fig 4A and Dataset EV5). A subset of 33 proteins differed significantly in G2019S carriers in both cohorts and all these proteins were upregulated in pathogenic *LRRK2* carriers. A pairwise comparison of the four subject groups (HC, NMC, iPD, and LRRK2 PD) using a Student *t*-test confirmed that the abundance of the overlapping proteins changed in a G2019S-dependent manner but was unaffected by the PD status (Fig 4B).

In total, 237 *LRRK2* status-associated proteins were quantified in both cohorts and the fold-changes of these were similar between the two cohorts (Pearson correlation: 0.34, Fig 4C). The lower correlation between the studies compared to the PD-associated proteins could be explained by the different distribution of PD$^+$ and PD$^-$ individuals in the two cohorts, which is only corrected for in the applied ANCOVA analysis but not in this Pearson correlation analysis. The effect sizes of the LRRK2-associated proteins were slightly larger for the Columbia cohort (Columbia: 1.43 (up) and 0.76 (down) vs. LCC: 1.39 (up) and 0.89 (down)). Interestingly, one of the proteins exhibiting the largest increase in *LRRK2* G2019S carriers in both cohorts was a phosphatase, the intestinal-type alkaline phosphatase (ALPI). As for the proteins that changed dependent on PD disease status, most *LRRK2* status-associated proteins were detected with at least two peptides and quantified with CVs below 50% (Appendix Fig S3B).

A GO term analysis revealed strong enrichment of proteins associated with lysosome-related terms such as "autolysosome", "lysosome", "lysosomal lumen", "azurophil granule lumen" and "lysosomal membrane" as well as "glycosphingolipid metabolic processes" in *LRRK2* G2019S carriers in both cohorts (Fig 4D). Among the proteins associated with the lysosome-related GO terms were multiple members of the cathepsin family including cathepsins A, B, C, D, H, L, O, S, and Z. The widely used lysosomal marker proteins, LAMP1 and LAMP2, were also significantly altered in *LRRK2* carriers in the LCC cohort, while LAMP3 was significantly changed in the Columbia cohort. In total, 13 proteins were associated with the GO term "sphingolipid metabolic process", most of them upregulated in *LRRK2* G2019S carriers. Among them were multiple lysosomal enzymes including GCase (encoded by *GBA*), galactocerebrosidase (GALC), sphingomyelin phosphodiesterase (SMPD1), and the beta-hexosaminidase subunits alpha and beta (HEXA and HEXB).

Heterozygous pathogenic mutations in *GBA* are one of the most common PD risk factors while homozygous loss of function mutations of the same protein causes the lysosomal storage disorder (LSD) Gaucher's disease. Past studies have described increased GCase activity in *LRRK2*-deficient mice and decreased GBA activity in *LRRK2* G2019S carrier neurons (Ferrazza *et al*, 2016; Ysselstein *et al*, 2019). Additionally, it has been reported that PD patients with mutations in both proteins develop symptoms at a younger age compared to patients with only one affected gene (Duran *et al*, 2013; Spitz *et al*, 2015; Yahalom *et al*, 2019). However, despite these reports, it remained unclear whether mutations in *GBA* and *LRRK2* contribute to the pathogenesis of PD via common pathways. Participants in the Columbia cohort were sequenced for mutations in *GBA* (Alcalay *et al*, 2015) and 22 individuals were found to carry a pathogenic mutation in this locus. To determine which proteins were changed specifically in carriers of *GBA* variants, we performed an ANCOVA analysis with sex, age at sample collection, PD status, and *LRRK2* status as confounding factors. Using an FDR of 5%, we found that levels of 74 proteins were affected by *GBA* (Fig 4E and Dataset EV5). Interestingly, only Intercellular adhesion molecule 1 (ICAM1), Adenosylhomocysteinase (AHCY), and Stomatin (STOM) were affected by pathogenic mutations in both *LRRK2* and *GBA*, suggesting that the two mutations largely affect distinct pathways. Furthermore, the *GBA*- and *LRRK2*-dependent protein fold-changes were poorly correlated (Pearson *r* = 0.21) (Fig 4F) but future well-powered studies on *GBA* cohorts are needed to firm up the data. Of note, most proteins associated with the GO term "glycosphingolipid metabolic process" were increased in *LRRK2* G2019S carriers but decreased in pathogenic *GBA* carriers, most notably GM2 activator (GM2A).

Together, we identified pathogenic mutant *LRRK2*-dependent protein signatures with a high correlation between the two independent cohorts. The *LRRK2* mutational status-dependent changes of

**Figure 4. Pathogenic *LRRK2*-dependent lysosomal dysregulation is reflected in the urinary proteome and distinct from pathogenic *GBA*-induced alterations.**

A   Proteins that differ significantly between pathogenic *LRRK2* carriers and controls using an ANCOVA analysis with sex, age, PD status and *GBA* status as confounders and an FDR of 5%.

B   Mean fold-changes for each of the 33 proteins that were *LRRK2*-dependently regulated in both cohorts using a pairwise t-test comparing the four subgroups (HC, NMC, iPD and *LRRK2* PD).

C   Correlation of mean fold-changes of the proteins changing *LRRK2*-dependently in the Columbia and LCC cohorts. Only proteins identified in both cohorts are shown (*n* = 237). The colors match to the GO terms shown in (D). All proteins associated with the GO term "glycosphingolipid metabolic process" labeled in purple are also associated with the lysosomal-related GO terms. Proteins overlapping between the two cohorts are labeled with their name.

D   Fisher exact test was performed to identify significantly enriched GO terms in the *LRRK2*-dependently regulated proteins in urine. All GO terms that were significant in either cohort are displayed (FDR < 5%).

E   Proteins that differ significantly between pathogenic *GBA* carriers and controls or between pathogenic *LRRK2* carriers and controls using an ANCOVA analysis with sex, age, PD status and *LRRK2/GBA* status as confounders and an FDR of 5%. In total, 237 proteins were differentially expressed in these two comparisons with 166 and 74 regulated proteins in the *LRRK2* carriers and *GBA* carriers, respectively, only three of which were common between both mutations.

F   Correlation of mean fold-changes of the proteins changing *LRRK2*-dependently and *GBA*-dependently (*n* = 237) in the Columbia cohort. Carriers of pathogenic variants in both *GBA* and *LRRK2* were excluded from the analysis. The colors match to the GO terms shown in (D).

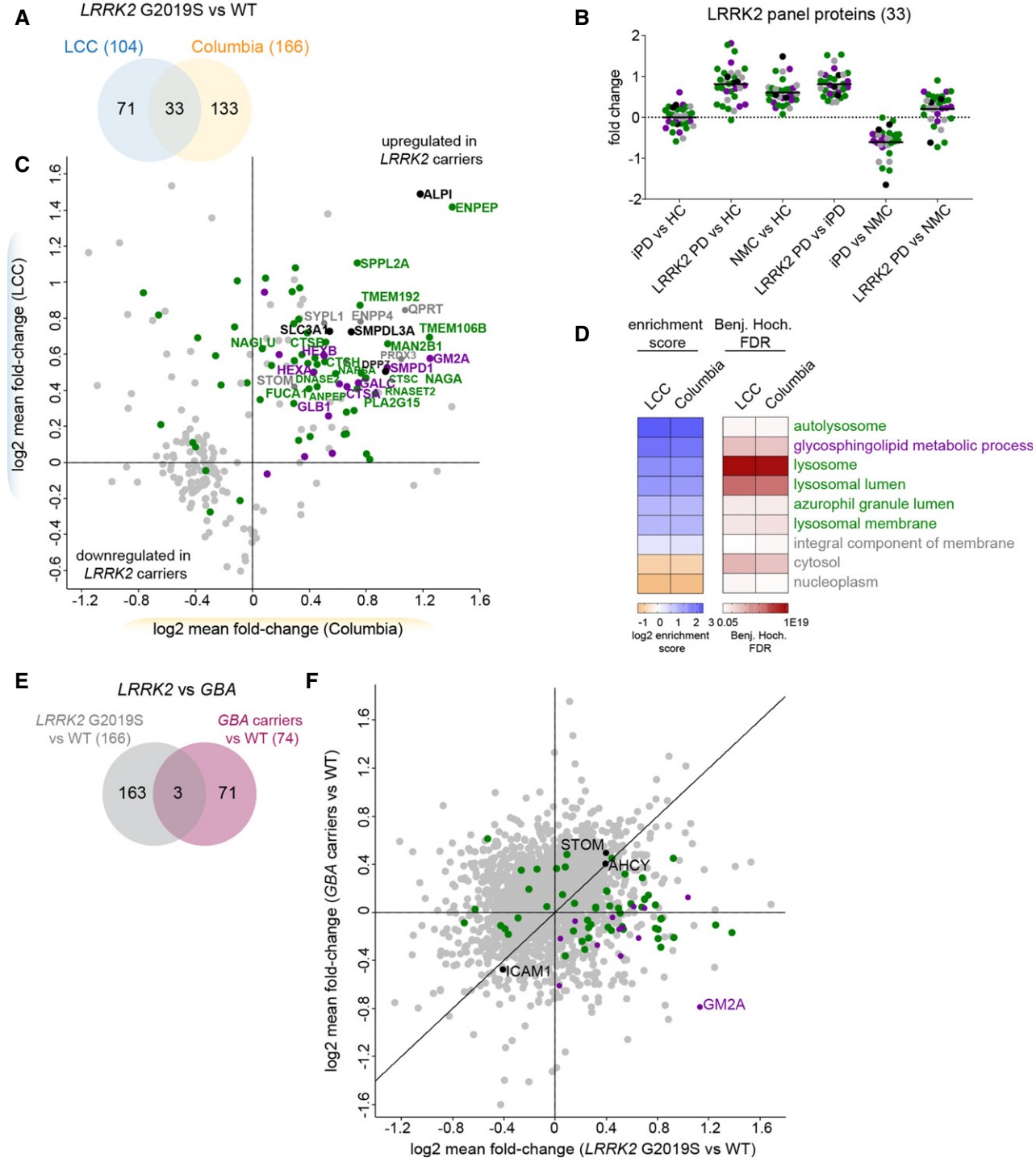

Figure 4.

the urinary proteome include lysosomal proteins that could serve as biomarkers to stratify subjects with pathogenic *LRRK2*.

### Correlation of proteome profiles with clinical parameters

Given that clinical parameters, including disease severity scores, were available for the Columbia cohort, we were interested in exploring whether any of these clinical parameters correlate with

proteomic changes we detected. We were especially interested in the cognitive capabilities of the participants as evaluated using the Montreal Cognitive Assessment (MoCA) test, and the motor performance as assessed using the Unified Parkinson's Disease Rating Scale part III (UPDRS-III). Within the Columbia cohort, MoCA scores ranged from 8 to 30, on a scale from 0, for most severe cognitive impairment, to 30, for no measurable cognitive impairment. We observed that two proteins, Tenascin-R (TNR) and Furin (FURIN),

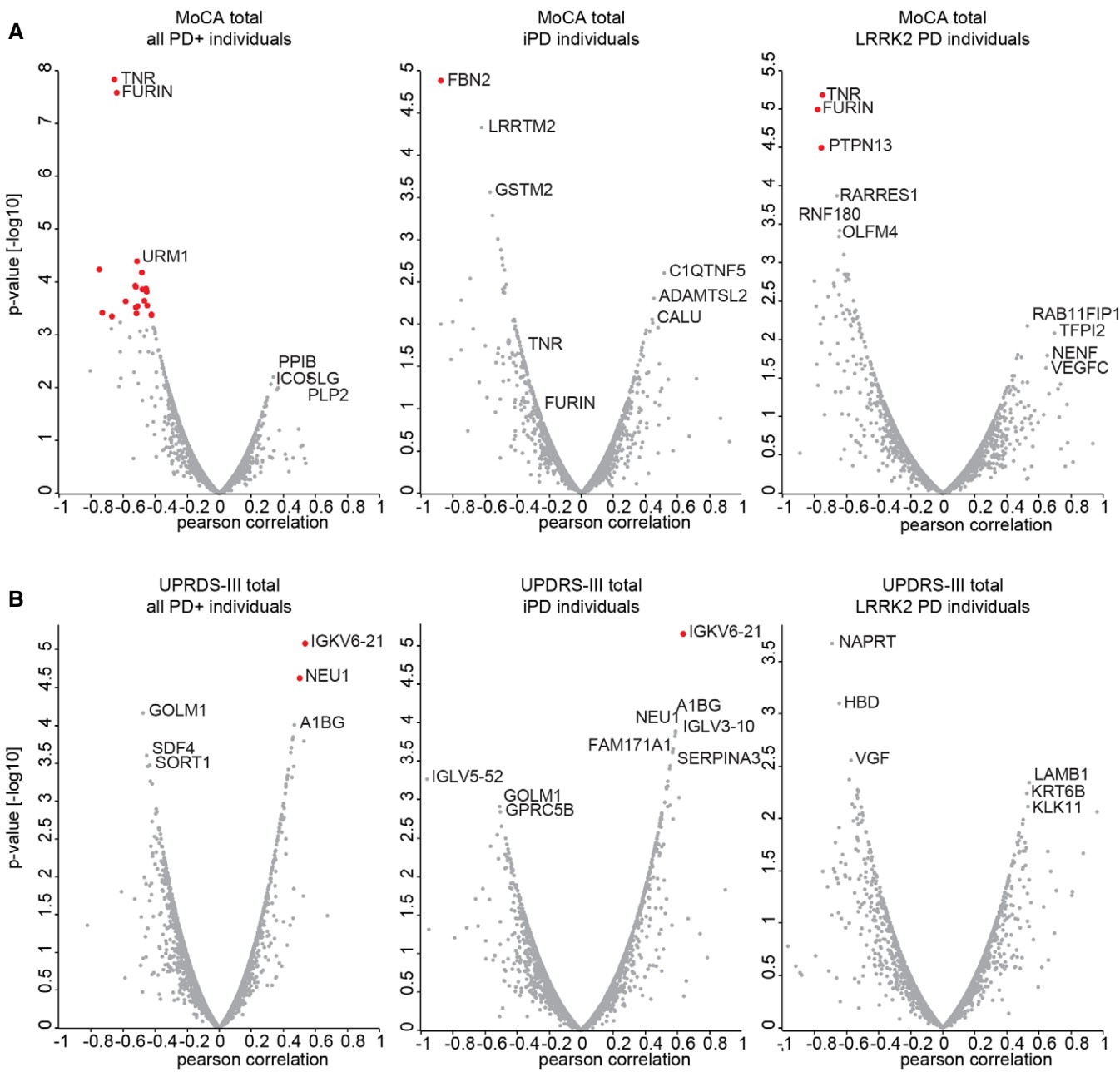

**Figure 5. Correlations with clinical parameters.**

A Pearson correlation scores and associated *P*-values [−log₁₀] of all protein intensities with the MoCA total score. Either all PD patients (left), iPD patients (middle) or LRRK2 PD patients (right) were included in the analysis. Significantly correlated proteins with an FDR of 5% after Benjamini–Hochberg correction are highlighted in red.

B Pearson correlation scores and associated *P*-values [−log₁₀] of all protein intensities with the UPDRS-III score. Either all PD patients (left), iPD patients (middle) or LRRK2 PD patients (right) were included. Significantly correlated proteins with an FDR of 5% after Benjamini–Hochberg correction are highlighted in red.

showed a strong negative correlation with the MoCA score in PD patients (TNR Pearson $r$: −0.66; FURIN $r$: −0.65; $P < 10^{-7}$ for both), mainly driven by *LRRK2* G2019S carriers (TNR $r$: −0.77; FURIN $r$: −0.78; $P < 10^{-5}$ for both) (Fig 5A and Appendix Fig S4). Interestingly, neither TNR nor FURIN was correlated with the age at sample collection in PD patients (TNR Pearson r: 0.19; FURIN r: 0.16). Furthermore, neither of the two proteins was significantly regulated

in the urinary proteomes of PD patients. When similar type of analysis was done with UPDRS-III scores, which ranged from 0 to 38 in the Columbia cohort (on a scale that may range from 0 for normal to a maximum possible score of 108 for most severely affected motor function), we observed that immunoglobulin kappa variable 6-21 (IGKV6-21), was the highest correlated protein in PD patients ($r$: 0.54, $P < 10^{-5}$; Fig 5B). This protein also exhibited one of the

highest fold-changes in abundance when comparing urine of PD patients with non-diseased individuals (Fig 3A). Of note, the correlation between UPDRS-III scores and levels of IGKV6-21 was mainly driven by iPD patients ($r$: 0.68; $P < 10^{-5}$) and much weaker in *LRRK2* G2019S PD patients ($r$: 0.36; not significant; Fig 5B). All correlation scores, $P$- and $q$-values can be found in Dataset EV6. Collectively, this analysis suggests that iPD and *LRRK2* G2019S patients could be stratified based on the differences between MoCA and UPDRS-III score correlations with different urine proteins.

### Machine learning-based classification of urinary proteomes

Finally, we assessed how well machine learning models can discriminate between PD patients and non-diseased individuals, between *LRRK2* G2019S and wild-type allele carriers, and between NMC individuals and *LRRK2* G2019S patients based on the acquired urinary proteome profiles. Since the accuracy of the model largely depends on the number of samples, we combined all samples from the Columbia and LCC cohorts for these analyses. We first selected and ranked which protein features to use in the machine learning model by employing a decision tree. To classify individuals as having PD or not, the decision tree selected the 15 most important features of the PD$^-$ vs. PD$^+$ urinary proteomes, with the intensity of PPIB, one of the proteins that displayed the largest difference in abundance when PD samples were compared to the controls (Fig 3A), being on top of the list (Appendix Fig S5A). Using these proteins, we trained an XGBoost model, a commonly applied algorithm for gradient boosting, a machine learning technique that is used to build robust predictive models based on ensembles of weaker predictions, such as decision trees. Samples were cross-validated by applying a stratified 4-fold split. This was repeated ($n = 15$) with shuffling the dataset to have a total of 60 train/test splits to achieve a robust estimate of model performance. Each time, we determined a receiver operating characteristic (ROC) curve and found the mean area under the curve (AUC), which is often used to assess the performance of a model, to be $0.84 \pm 0.05$ (Fig 6A). On average, we correctly classified 91 out of 117 PD patients and 77 out of 106 controls in the test sets (Fig 6B). Accordingly, the machine learning model reached a sensitivity of 78% and a specificity of 73%. When we trained the model on the one cohort and tested it on the other cohort, we obtained AUCs of 0.86 or 0.72, further demonstrating the robustness of the model (Fig 6C).

We next used the same machine learning methods to classify G2019S and wild-type *LRRK2* carriers using the same strategy as described above. The decision tree selected the 15 most important features, with the intensity of ENPEP being the most important one (Fig 4A; Appendix Fig S5B). Using these proteins and the XGBoost algorithm, we obtained a mean AUC of the ROC curves of $0.87 \pm 0.04$ (Fig 6D). For the test sets, we could correctly classify 73 out of the 99 *LRRK2* G2019S carriers and 103 out of the 123 wild-type allele carriers, corresponding to a 74% sensitivity and 84% specificity (Fig 6E). When we trained the model on one of the cohorts and tested it on the other, we obtained AUCs of 0.76 or 0.80 (Fig 6F). Additionally, we trained the model on all individuals with a known *LRRK2* status and classified the sample from the Columbia cohort with an unknown *LRRK2* status with an 87% probability to be wild-type *LRRK2*. After we had finished this machine learning modeling, the mutational status of this individual

was determined as wild-type *LRRK2*, further verifying the machine learning model.

Encouraged by these results, we wanted to see how well machine learning can discriminate *LRRK2*$^+$ PD patients from NMCs that also carry a *LRRK2* mutation and are at increased risk of developing the disease. Using a decision tree, we selected seven proteins for training the model (Appendix Fig S5C). Interestingly, VGF, a neurotrophic factor, was identified as the most important feature. When using these features to train a classifier with our cross-validation scheme, the obtained mean AUC of the ROC curve was $0.94 \pm 0.05$ and the obtained sensitivity and specificity were both 88% (Fig 6G and H). Using samples from only one cohort as a training set and applying the model to the other cohort resulted in AUCs of 0.93 and 0.74 (Fig 6I). Taken together, machine learning allowed us to classify the PD and *LRRK2* states with high specificities and sensitivities.

## Discussion

The pathophysiology of PD leads to progressive decline of motor function and results in numerous quality of life issues for patients and their families, and inevitably leads to death within 7 to 14 years from the initial diagnosis. The majority of previous PD biomarker discovery and validation efforts have focused on CSF, serum, and blood (Chen-Plotkin *et al*, 2018). Additional strategies included targeted monitoring of α-synuclein levels, given the known relationship between α-synuclein accumulation and PD progression (Fields *et al*, 2019). To address this problem, we developed a shotgun proteomics workflow for urinary proteome profiling. We chose to focus on urine given the non-invasive nature of obtaining clinical samples, which is a major advantage when developing a strategy that can be used not only for diagnostic and prognostic purposes, but for long-term disease progression and treatment response monitoring. Additionally, instead of focusing on a single biomarker and/ or a subset of molecular entities, our shotgun proteomics approach provides a multiparameter global map of the disease state. We previously showed that this strategy can yield powerful, data-driven descriptors of a disease (Geyer *et al*, 2016b; Niu *et al*, 2019; Bader *et al*, 2020), and we now confirm, for the first time, that this also works for urinary proteome analysis in the context of a complex neurodegenerative disease, such as PD.

Our quantitative shotgun proteomics workflow represents a sensitive and scalable approach for rapid analysis of a large number of samples. Applying this workflow to more than 200 urine samples from two independent cohorts allowed us to precisely quantify on average more than 2,000 proteins per sample while using minimal sample amounts of less than 100 μl. Our approach successfully determined proteins with abundances that varied over more than five orders of magnitude and quantified more than 1,200 proteins with a CV below 20% across the two cohorts, highlighting the high depth and precision of our study. Moreover, the observed variability between samples was much smaller than the biological variability between subjects, further illustrating the quantitative robustness of our workflow.

Another factor contributing to the quality of the urinary proteome dataset reported here is the composition of the cohorts we analyzed. The cohorts included two types of controls, the healthy controls as well as asymptomatic individuals that are carriers of PD-

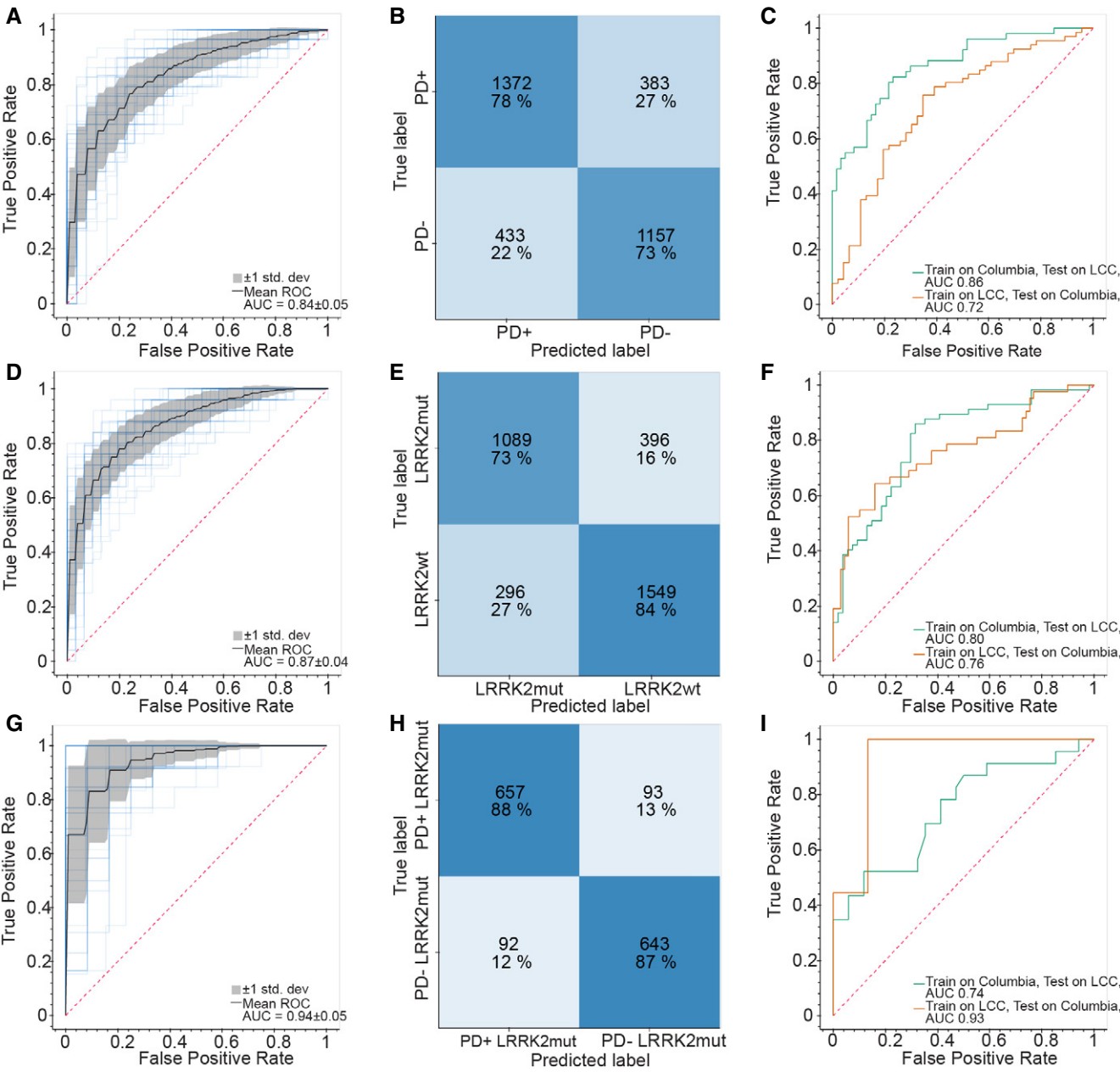

**Figure 6. Machine learning-based classification of PD and *LRRK2* status.**

A   Receiver operating characteristic (ROC) curve for the XGBoost-based model to classify PD+ vs. PD− individuals. Random performance is indicated by the dotted diagonal line. The gray area represents the standard deviation from the mean ROC curve. The blue lines show the values for a total of 15 repeats with four stratified train-test splits.

B   Confusion matrix showing the model performance for classifying PD+ vs. PD− individuals. Numbers represent the mean number from 15 repeats of cross-validation with four stratified train-test splits.

C   ROC curve for the XGBoost-based model when trained on one cohort and tested on the other cohort. Random performance is indicated by the dotted diagonal line.

D   Same as (A) but for classification of *LRRK2* G2019S vs. *LRRK2* WT carriers.

E   Same as (B) but for classification of *LRRK2* G2019S vs. *LRRK2* WT carriers.

F   Same as (C) but for classification of *LRRK2* G2019S vs. *LRRK2* WT carriers.

G   Same as (A) but for classification of PD+ vs. PD− in *LRRK2* G2019S carriers.

H   Same as (B) but for classification of PD+ vs. PD− in *LRRK2* G2019S carriers.

I   Same as (C) but for classification of PD+ vs. PD− in *LRRK2* G2019S carriers.

associated mutation G2019S *LRRK2*. The cohorts also included PD patients with and without the mutation, and patients of both sexes, thus allowing for different types of comparisons. For example, the global correlation map and PCA analysis showed that the sex of an individual has a dominant effect on the urinary proteome. This is in line with basic physiology and previous reports (Shao *et al*, 2019) but highlights the importance of incorporating sex as a confounding factor for statistical analyses. This is further illustrated by the fact that 42 and 12% of the proteins with differential abundance in PD patients vs. controls, as well as 35 and 14% of the proteins that exhibit different abundance in a *LRRK2* mutational status-dependent manner also significantly differed between the sexes in the Columbia and LCC cohorts, respectively.

Applying our "rectangular" strategy for biomarker discovery (Geyer *et al*, 2017), we discovered 361 and 237 significantly altered proteins in PD patients and pathogenic *LRRK2* carriers, respectively. The observed overlap of proteins exhibiting significantly perturbed levels in the two independent cohorts confirms that valuable information can be inferred from the urinary proteome for neurodegenerative diseases. The limited number of samples for each of the four subject groups as well as demographic and ethnic heterogeneity between them are limitations of the present study. However, we note the scalability of our workflow, which will allow its application to larger cohorts with more comprehensive genetic and clinical information. This extension of our work will be important to further validate our results and to discover additional biomarker candidates with improved statistical power. Building up on our data and further validation studies, specific targeted clinical assays could be developed for clinical use (Karayel *et al*, 2020).

Our data analysis led to several interesting observations that might suggest opportunities for follow up. Here, we will briefly discuss only a small number of such examples. For those interested in more in-depth data mining, we made our datasets available via a publicly accessible depository (see Materials and Methods for accession numbers). An interesting insight that emerged from the GO term analysis of PD patient vs. control proteomes is that proteins associated with "bone development" are regulated in PD patients. The enzyme PPIB was significantly upregulated in PD patients in both cohorts. This cyclophilin assists the folding of type I collagen and can protect cells against $MPP^+$-induced cell death in a PD cell culture model (Oh *et al*, 2016). Inhibitors of the closely related family member cyclophilin D (CypD) are considered as therapeutic agents against several neurodegenerative diseases including PD (Fayaz *et al*, 2015). Most other proteins associated with the GO term "bone development" were downregulated in PD patients, in line with recent findings that PD patients frequently suffer from osteoporosis and osteopenia (Torsney *et al*, 2014; Handa *et al*, 2019). Going forward, it would be important to examine the relationship between PD progression and bone health more closely, as this connection is currently underexplored.

Another enriched term was "growth factor activity", although none of the proteins associated with this term was significantly regulated in both cohorts. Growth factors and particularly neurotrophic factors have gained strong interest as therapeutic agents in Parkinson's disease but so far have not produced convincing clinical benefits (Paul & Sullivan, 2019). The neurosecretory protein VGF was strongly decreased in PD patients in both cohorts (Columbia: 0.24, LCC: 0.54) but only reached statistical significance in the

better-controlled Columbia cohort. VGF is synthesized as a prohormone and proteolytically processed to various biologically active peptides. In this study, we identified peptides covering most of the VGF sequence, including sequences contained in the neuroendocrine regulatory peptide-1. However, the applied tryptic digestion complicates a direct link to the endogenous hormone peptides. VGF is exclusively synthesized and secreted by neuronal and neuroendocrine tissues. In the CNS, VGF promotes neurite growth and exhibits neuroprotective activity, while it also regulates energy homeostasis in peripheral tissues. Gene expression of VGF in the cortex (Henderson-Smith *et al*, 2016) and peptides derived from this gene are reduced in post-mortem parietal brain cortex and plasma from PD patients (Cocco *et al*, 2010; Cocco *et al*, 2020). Furthermore, VGF has been suggested as a biomarker in CSF for Alzheimer's disease (AD) and Amyotrophic lateral sclerosis (ALS) and its expression was reduced in the CSF of AD and ALS patients compared to controls (Carrette *et al*, 2003; Pasinetti *et al*, 2006).

We also identified several apolipoproteins - the major proteinaceous constituent of lipoproteins - to be significantly upregulated in PD patients in the Columbia cohort. They have been linked to neurodegenerative disorders including Alzheimer disease, including in our recent proteomic study of CSF (Bader *et al*, 2020). APOE variants were shown to exhibit neuroprotective activity (reviewed in (Emamzadeh, 2017)). APOA1 is the major protein component of plasma high-density lipoprotein and its low levels in CSF and plasma have been reported as a potential PD biomarker (Wang *et al*, 2010; Qiang *et al*, 2013; Swanson *et al*, 2015). While APOE and ApoA1 are the most abundant apolipoproteins in the CSF and highly enriched in the brain (Borghini *et al*, 1995; Koch *et al*, 2001), APOC1—a less-abundant brain apolipoprotein—was implicated in Alzheimer disease although its regulation and possible role is poorly understood (Petit-Turcotte *et al*, 2001).

In another illustrative example, we analyzed proteomic differences between patients with and without a major inherited mutation associated with familial PD: *LRRK2* G2019S. Lysosomal dysregulation and associated α-synuclein aggregation appear to be a central event in the pathogenesis of PD (Alessi & Sammler, 2018) and LRRK2, through its regulation of the endolysosomal pathways, is a key player in this mechanism (Henry *et al*, 2015; Roosen & Cookson, 2016; Vidyadhara *et al*, 2019; Kuwahara & Iwatsubo, 2020). Interestingly, the *LRRK2*-dependent signature in the urinary proteome seemed to be more consistent than the PD-dependent signature, as indicated by the larger overlap of 33 vs. 10 proteins between the two cohorts. This may also be explained by the fact that the disease diagnosis and thus the classification of individuals in $PD^-$ and $PD^+$ is based on multiple clinical criteria, resulting in more heterogeneous populations than classification in $LRRK2^-$ and $LRRK2^+$ based on sequencing of the locus. This suggests that the genetic mutation of *LRRK2* not only manifests in the central nervous systems but also dysregulates multiple pathways in distal organs such as the bladder and kidney, where LRRK2 is actually highly expressed (Biskup *et al*, 2007). Our results demonstrate that urine of pathogenic *LRRK2* carriers strongly reflects lysosomal dysregulation associated with increase in LRRK2 activity (Alessi & Sammler, 2018). These major proteome changes are in agreement with a previous study that observed more than 2,000 proteins to be affected by pathogenic LRRK2 (Connor-Robson *et al*, 2019). One of the strongest upregulated proteins in *LRRK2* G2019S carriers was

the alkaline phosphatase ALPI. We suggest that this enzyme may counteract the hyperactive kinase by as yet undiscovered feedback mechanisms. Interestingly, knockdown of ALPI has been shown to decrease both LRRK2 levels and activity in cells (Berndsen *et al*, 2019). We also found several lysosomal proteins including α- and β-subunit of β-hexosaminidase A (HEXA and HEXB), GM2A, and GCase, whose genes are associated with LSDs, to be upregulated in *LRRK2* G2019S carriers in both cohorts. Mutations in many LSD genes have been associated with PD, suggesting common pathogenic mechanisms underlie both diseases. The GO term "glycosphingolipid metabolic process" was enriched among *LRRK2*-regulated proteins, in agreement with increased interest in understanding how sphingolipids contribute to PD that stems from the fact that several PD-associated genes including *GBA* are linked to their metabolism (Lin *et al*, 2019; Plotegher *et al*, 2019). Ceramide levels are increased in *LRRK2*-deficient mouse brains and this decrease is regulated by *GBA* (Ferrazza *et al*, 2016). It still remains unclear how the disruption of sphingolipid metabolism may result in PD-associated neurodegeneration or if LRRK2 directly or indirectly regulates this process. Our data suggest that pathogenic mutations in *GBA* and *LRRK2* mainly affect distinct regulatory networks, as only three proteins were significantly altered in common by mutations in both genes. However, further studies on larger *GBA* cohorts are needed to confirm and extend our findings.

One of the cohorts we analyzed (Columbia) included information on clinical scores of cognitive and motor performances. This allowed us to correlate proteomic changes to clinical score, thus revealing that TNR and FURIN levels were strongly correlated with higher cognitive impairment. FURIN is a protease and involved in NMDA-induced neuronal injury (Yamada *et al*, 2018). Furthermore, its homologue in the fruit fly, Furin1, has been reported to be a translational target of pathogenic LRRK2 and to be involved in neurotoxicity (Maksoud *et al*, 2019). TNR is a neural extracellular matrix protein exclusively expressed in the brain. It is involved in neurogenesis (Xu *et al*, 2014) and extracellular matrix aggregates in the brain called perineural nets (Morawski *et al*, 2014). Of note, rare TNR variants have also been associated with familial PD (Farlow *et al*, 2016). Interestingly, IGKV6-21 was highly upregulated in PD patients and also strongly correlated with the UPDRS-III score. Although the underlying biology is unclear, the association with both PD risk and severity makes this V region a promising biomarker candidate to pursue in future studies.

To extend utility of our datasets, we developed a machine learning model for stratifying PD patients and *LRRK2* G2019S carriers with high sensitivities and specificities. Importantly, the machine learning model excelled in classifying the PD status in *LRRK2* G2019S carriers. This is of high interest, because although these carriers are at an increased risk of developing PD, there is no predictive marker to determine whether or not and when a mutation carrier develops the disease. Given the performance of the machine learning model, VGF, LTF, CELA3A, TUBB4B, and SOD2 are promising candidates as predictive markers to early indicate disease development.

In summary, we have demonstrated that a distal body fluid like urine contains brain-specific proteins and can inform about the disease and mutation status in a neurodegenerative disease. Our urinary proteomics workflow is relatively straightforward, readily scalable, and thus easily applicable to larger and more powerful cohorts. It would be important to also apply it to longitudinal data to confirm increased levels of PPIB and IGKV6-21 in PD patients and

VGF as a potential indicator for disease manifestation in *LRRK2* G2019S carriers but also identify new biomarkers for PD risk and disease progression in idiopathic and genetic forms of PD. Our results demonstrate that urinary proteome profiling enables the discovery of better biomarkers, which could have a major impact on important aspects of disease management: (i) a diagnostic biomarker will enable early and objective diagnosis of PD, (ii) a prognostic biomarker will provide information about the progression of the disease, and (iii) predictive and treatment response biomarkers will allow to monitor whether and how the patients respond to a therapy. Reliable biomarkers assessing LRRK2 activity can also aid with monitoring compliance of LRRK2 kinase inhibitors and treatment efficacy, early detection of non-manifesting carriers to prevent disease onset and stratify idiopathic PD patients who could benefit from LRRK2-based therapies.

# Materials and Methods

### Study cohorts

In this study, urine samples from two independent cross-sectional cohorts were analyzed. Both studies were approved by local institutional review boards, and each participant signed an informed consent. All experiments conformed to the principles set out in the WMA Declaration of Helsinki and the Department of Health and Human Services Belmont Report.

The first cohort was recruited at Columbia University Irving Medical Center (Columbia cohort) and its participants donated urine under a MJFF-funded *LRRK2* biomarker project from March 2016 to April 2017. This cohort contained 35 healthy individuals without pathogenic LRRK2 mutation (HC), 16 non-manifesting carriers of the LRRK2 G2019S mutation (NMC), 40 idiopathic PD patients without pathogenic LRRK2 mutation (iPD) and 28 PD patients with the pathogenic *LRRK2* G2019S mutation (LRRK2 PD) and 1 PD patient with an unknown mutation status of *LRRK2*. Motor performance was evaluated using the Unified Parkinson's Disease Rating Scale part III (UPDRS-III), and cognitive functioning was assessed using the Montreal Cognitive Assessment (MoCA) test. Genotyping for *LRRK2* G2019S and *GBA* mutations was conducted as previously described (Alcalay *et al*, 2015).

To confirm findings from the Columbia cohort, urine from a second cohort consisting of 115 biobanked urine samples from the Michael J. Fox Foundation for Parkinson's Research (MJFF)-funded *LRRK2* Cohort Consortium (LCC) was analyzed. The cohort used in this study was an exploratory subset of a larger cohort and contained 26 healthy individuals without pathogenic *LRRK2* mutation (HC), 37 non-manifesting carriers of the *LRRK2* G2019S mutation (NMC), 29 idiopathic PD patients without pathogenic *LRRK2* mutation (iPD), and 23 PD patients with the pathogenic *LRRK2* G2019S mutation. UPDRS-III and MoCA scores were not available for subjects from the LCC cohort.

### Quality assessment

To generate the urine-specific quality marker panel, we recruited three volunteers from within the Department of Proteomics and Signal Transduction at the Max Planck Institute of Biochemistry

who kindly donated 10 ml of urine at three different time points during a day and provided a written informed consent, with prior approval of the ethics committee of the Max Planck Society.

Following the collection, urinary samples were centrifuged at 2000g for 10 min, supernatants were harvested, and pellets were resuspended in 100 μl of urea sample solution. 100 μl of each supernatant and the entire 100 μl of the resuspended pellets were used for sample preparation as described below. A sample was flagged for potential contamination if the summed intensity of all proteins in the respective quality marker panel differed more than 2 standard deviations from the mean of all samples within the cohort.

## Sample preparation

The undiluted neat urine as well as the cleared and pelleted urine samples for the urine-specific quality marker panel were prepared using MStern Blot protocol as described previously (Berger *et al*, 2015). Briefly, 100 μl of urine was first diluted in 300 μl of urea sample solution (8 M urea in 50 mM ammonium bicarbonate (ABC)) and subsequently mixed with 30 μl of 150 mM dithiothreitol (DTT) solution (150 mM DTT, 8 M urea, 50 mM ABC) in a 96-well plate. The resulting solution was incubated for 20 min at room temperature. Reduced cysteine side chains were alkylated by adding 30 μL of iodoacetamide (IAA) solution (700 mM IAA, 8 M urea, 50 mM ABC) and incubated for 20 min in the dark. During incubation, each well of the 96-well PVDF membrane plates (MSIPS4510, Merck Millipore) was activated and equilibrated with 150 μl of 70% ethanol/water and urea sample solution, respectively. The urine samples were transferred through the PVDF membranes using a vacuum manifold (MSVMHTS00, Merck Millipore). Adsorbed proteins were washed two times with 150 μl of 50 mm ABC. Digestion was performed at 37°C for 2 hours by adding 100 μl digestion buffer (5% v/v acetonitrile (ACN)/50 mm ABC) containing 0.35 μg per well of each protease trypsin and LysC. After incubation in a humidified incubator, the resulting peptides were collected by applying vacuum and remaining peptides were eluted twice with 75 μl of 40%/0.1%/59.9% (v/v) acetonitrile/formic acid/water. The pooled peptide solutions were dried in a vacuum centrifuge.

Peptides resuspended in 0.1% trifluoroacetic acid (TFA) were desalted on C18 StageTips as described in (Kulak *et al*, 2014). The StageTips were centrifuged at 1,000g for washing with 0.1% TFA and elution with 80% ACN/0.1% TFA. The eluate was evaporated to dryness using a vacuum centrifuge and peptides were resuspended in 10μl buffer A* (2% ACN/0.1% TFA and stored at −20°C. Samples were thawed shortly before mass spectrometric analysis and shaken for 2 min at 2,000 rpm (thermomixer C, Eppendorf). Peptide concentrations were measured optically at 280 nm (Nanodrop 2000, Thermo Scientific) and subsequently equalized using buffer A*. 500ng peptide was subjected to LC-MS/MS analysis.

Cohort-specific libraries for data-independent analyses were generated by pooling of 25 randomly selected samples of each cohort. Sample pools were fractionated into 24 fractions each by high pH (pH 10) reversed-phase chromatography as described earlier (Kulak *et al*, 2017). Fractions were concatenated automatically by shifting the collection tube every 120 s and subsequently dried in a vacuum centrifuge and resuspended in buffer A*.

To increase the depth of each library, extracellular vesicles (EV) were isolated from pooled urine samples of each cohort by ultra-

centrifugation as described earlier(Fraser *et al*, 2016). Briefly, 8.5 ml of 6 urine samples per group (*LRRK2⁻*/PD⁻, *LRRK2⁺*/PD⁻, *LRRK2⁻*/PD⁺, and *LRRK2⁺*/PD⁺) were pooled and centrifuged at 10,000 *g* for 30 min at 4°C and supernatant was transferred and then centrifuged again at 100,000 *g* for 1 h at 4°C. Supernatants were discarded and pellets were washed by adding 30 ml PBS and centrifugation at 100,000 *g* for 1 h at 4°C. Supernatant was discarded and pellets were resuspended in 100 μl of a sodium deoxycholate-based lysis buffer containing chloroacetamide (PreOmics GmbH) and heated to 95°C for 10 min for reduction and alkylation. After cooling to room temperature, 0.75 μg of each protease trypsin and LysC were added to each sample and digestion was performed at 37°C overnight. Peptides were desalted with SDB-RPS (styrenedivinylbenzene-reverse phase sulfonate) StageTips. Samples were mixed with 5 volumes of 1% TFA/isopropanol for loading on StageTips and subsequently washed once with 1%TFA/isopropanol and once with 0.2% TFA as described earlier (Kulak *et al*, 2014). Peptides were eluted using 80%/5% ACN/ammonium hydroxide. The eluate was completely dried using a vacuum centrifuge and resuspended in 0.1% formic acid. Peptides were then separated into 8 fractions by high pH reversed-phase chromatography as described above for the libraries.

To determine coefficients of variation for the Columbia cohort, urine from five donors in triplicates on one plate were subjected to sample preparation (intra-plate) and this was repeated on three different plates (inter-plate). For the LCC cohort, urine from three donors in duplicates on one plate was subjected to sample preparation (intra-plate). Urine from nine other subjects were prepared on two different plates (inter-plate).

## LC-MS/MS analysis

LC-MS/MS analysis was performed on an EASY-nLC 1,200 coupled to a Q Exactive HF-X Orbitrap mass spectrometer via a nano-electrospray ion source (all Thermo Fisher Scientific). Purified peptides were separated at 60 °C on 50cm columns with an inner diameter of 75μm packed in-house with ReproSil-Pur C18-AQ 1.9μm resin (Dr.Maisch GmbH). Mobile phases A and B were 99.9/0.1% water/ formic acid (v/v) and 80/20/0.1% acetonitrile/water/formic acid (v/v/v). For the LCC cohort, the flow rate was constant at 300 nl/ min and the initial concentration of 5% B was linearly increased to 30% B within 36 min, and then increased further to 95% within 6 min with a 3 min plateau at the end. For the Columbia cohort, the flow rate was constant at 350 nl/min and the initial concentration of 5% B was linearly increased to 30% B within 35 min, and then increased further to 95% within 5 min with a 5 min plateau at the end.

MS data were acquired in the data-independent acquisition (DIA) scan mode for single-shot patient samples, using the MaxQuant Live software and spectral processing with phase-constrained spectrum deconvolution (phi-SDM; Grinfeld *et al*, 2017; Wichmann *et al*, 2019). Full MS scans were acquired in the range of m/z 300–1,650 at a resolution of 60,000 at m/z 200 and the automatic gain control (AGC) set to 3e6. For the Columbia cohort, additionally two BoxCar scans with 12 isolation windows each and a resolution of 60,000 at m/z 200 were acquired (Meier *et al*, 2018) . Full MS events were followed by 33 MS/MS windows (LCC cohort) or 58 MS/MS windows (Columbia cohort) per cycle in the range of m/z 300–

1,650 at a resolution of 15,000 at m/z 200. For the LCC cohort, higher-energy collisional dissociation MS/MS scans were acquired with a stepped normalized collision energy of 25/27.5/30 and ions were accumulated to reach an AGC target value of 3e6 or for a maximum of 30 ms. For the Columbia cohort, higher-energy collisional dissociation MS/MS scans were acquired with a normalized collision energy of 27 and ions were accumulated to reach an AGC target value of 3e6 or for a maximum of 22 ms.

All fractionated samples including EV fractions were acquired with a top 12 data-dependent acquisition (DDA) scan mode. Full MS scans were acquired in the range of m/z 300–1,650 at a resolution of 60,000 (Columbia cohort) or 120,000 (LCC cohort) at m/z 200. The automatic gain control (AGC) target was set to 3e6. Higher-energy collisional dissociation MS/MS scans were acquired with a normalized collision energy of 27 at a resolution of 15,000 at m/z 200. Precursor ions with charge states of 2-7 were isolated in a 1.4 Th window and accumulated to reach an AGC target value of 1e5 or for a maximum of 60 ms. Precursors were dynamically excluded for 20 s after the first fragmentation event.

**Mass spectrometry data processing**

The MS data of the fractionated pools (DDA MS data, 24 neat pool urine and 8 EV fractions) and the single-shot subject samples (DIA MS data, 165 and 132 samples in Columbia and LCC, respectively) were used to generate a DDA library and directDIA library, respectively, which were computationally merged into two cohort-specific hybrid libraries using Spectronaut version 13.9.191106.43655 (Biognosys AG). For all experiments except the machine learning, the two cohorts were quantified separately in Spectronaut. A minimum of 3 and a maximum of 10 fragments were required for each peptide in the library. The hybrid spectral libraries were subsequently used to search the MS data of the single-shot patient samples in the Spectronaut software. All searches were performed against the human SwissProt reference proteome of canonical and isoform sequences with 42,431 entries downloaded in July 2019. Searches used carbamidomethylation as fixed modification and acetylation of the protein N-terminus and oxidation of methionines as variable modifications. Trypsin/P proteolytic cleavage rule was used, permitting a maximum of 2 missed cleavages and a minimum peptide length of 7 amino acids. The Q-value cutoffs for both library generation and DIA analyses were set to 0.01. For generation of the global correlation map, the individual protein correlations with clinical parameters, and the machine learning, the Q-value data filtering setting in Spectronaut was set to "Qvalue" to use every peptide passing the Q-value threshold for the protein group quantification. For all other analyses, the setting was set to "Qvalue percentile" with a cutoff of 25%, to use only those peptides for the protein quantification that passed the Q-value threshold in at least 25% of all analyzed samples. The "Qvalue percentile" setting results in a complete data matrix with no missing values, as the noise is quantified and reported if the peptide did not pass the Q-value threshold.

**Bioinformatics data analysis**

The Perseus software package versions 1.6.0.7 and 1.6.1.3 and GraphPad Prism version 7.03 were used for the data analysis

### The paper explained

#### Problem

Parkinson's disease (PD) is the second most common neurodegenerative disease and due to the aging population, its prevalence is steadily increasing. Patients are typically diagnosed when motor impairment manifests and irreversible brain damage has already occurred. Development of disease-modifying therapeutics has been hampered by the lack of specific tests for early detection of PD.

#### Results

We developed a scalable, sensitive, and reproducible mass spectrometry-based workflow for urinary proteome profiling and applied it to two independent urine cohorts with PD patients and controls. The cohorts also contained carriers of the LRRK2 G2019S mutation, which is a strong risk factor for developing PD and frequently found in familial forms of the disease. In total, 361 proteins were significantly different in PD patients compared to non-diseased individuals, including proteins involved in protein folding and canonical ribonucleases. Individuals with the G2019S mutation showed 237 differentially expressed proteins and lysosomal dysregulation in urine. Machine learning successfully classified PD and LRRK2 status on the basis of the urinary proteome alone (specificities of 73% and 84% and sensitivities of 78% and 74%, respectively). Classification of manifesting and non-manifesting LRRK2 G2019S carriers reached a remarkable specificity and sensitivity of 88%.

#### Impact

These results demonstrate that urine is a valuable non-invasive body fluid for the detection of biomarkers for a neurodegenerative disease such as PD. Furthermore, this study improves our understanding of PD biology and identified pathways to be further investigated for the development of novel treatment strategies.

(Tyanova *et al*, 2016). Protein intensities were log2-transformed for further analysis apart from correlation and coefficient of variation analysis. Coefficients of variation (CVs) were calculated in Perseus for all inter-plate and intra-plate combinations of samples, the median values were reported as overall coefficient of variation. The protein CVs of the main study were calculated likewise within cohorts individually. The protein abundance levels were cross-correlated to generate a matrix of correlation coefficients. Unsupervised hierarchical clustering was performed using Perseus and proteins were clustered based on Pearson correlation scores. For generation of the abundance curves, median protein abundances across all samples within a proteome were used. ANCOVA analysis was performed in python (version 3.7.6) using the pandas (version 1.0.1), numpy (version 1.18.1), and pingouin (version 0.3.4) packages. For the ANCOVA analysis, age at sample collection, *LRRK2* status (only in PD$^+$ vs. PD$^-$), *GBA* status (only Columbia cohort LRRK2$^+$ vs. LRRK2$^-$), and PD status (only LRRK2$^+$ vs. LRRK2$^-$) were set as confounding factors. The FDR was set to 5% after Benjamini–Hochberg correction. GO annotations were matched to the proteome data based on Uniprot and Ensemble identifiers. Annotation term enrichment was performed with Fisher exact test in Perseus separately for each cohort. Annotation terms were filtered for terms with an FDR of 5% after Benjamini–Hochberg correction in each cohort. Calculation of Pearson correlation scores and associated *P*-values of protein intensities to UPDRS-III and MoCA scores was performed in Perseus. Benjamini–Hochberg

correction to calculate which proteins were significantly correlated with an FDR of 5% was performed in R using the "stats" package.

### Machine learning

Data processing and machine learning was performed in Python (version 3.7.3). Missing values were not imputed and protein intensities were normalized using the ScandardScaler method from the scikit-learn package (0.21.3). The XGBoost package (version 0.90) was used to classify the samples and results were plotted using the bokeh library (2.1.1). Features were selected using a decision tree. Samples from both Columbia and LCC cohorts were used for the model and cross-validated using four stratified training/test splits and 15 repeats were applied. To assess sensitivity and specificity of the model, the results of the test sets were summed and averaged from 15 repeats.

## Data availability

The mass spectrometry proteomics data have been deposited to the ProteomeXchange Consortium via the PRIDE partner repository with the dataset identifier PXD020722.

- Proteomics raw files: PRIDE PXD020722 (www.ebi.ac.uk/pride/archive/projects/PXD020722).

**Expanded View** for this article is available online.

### Acknowledgements
Biospecimens used in the analyses presented in this article were obtained from the MJFF-sponsored *LRRK2* Cohort Consortium (LCC). For up-to-date information on the study, visit www.michaeljfox.org/news/lrrk2-cohort-consortium. We thank all members of the Proteomics and Signal Transduction Group at the Max Planck Institute of Biochemistry and the Clinical Proteomics Group at the NNF Center for Protein Research for help and discussions and in particular Jakob Bader, Philipp Geyer, Igor Paron, Christian Deiml, and Alexander Strasser for helpful discussions and technical assistance. We further thank Hanno Steen, Dario Alessi, and Suzanne Pfeffer and their group members for helpful discussion. We thank employees of the Michael J. Fox Foundation for Parkinson's Research for helpful discussions. The work carried out in this project was supported by the Max Planck Society for the Advancement of Science and The Michael J. Fox Foundation. Open Access funding enabled and organized by Projekt DEAL.

### Author contributions
SVW and OK designed the experiments, performed, analyzed and interpreted all data. MTS helped with the machine learning. SP, MS, KM and RNA were responsible for sample collection and their distribution. SP, KM and RNA helped with interpretation of the results. MM supervised and guided the project, interpreted results and wrote the manuscript with SVW and OK.

### Conflicts of interest
The authors declare that they have no conflict of interest.

### For more information
i   www.michaeljfox.org
ii  www.michaeljfox.org/news/lrrk2-cohort-consortium
iii www.biochem.mpg.de/mann

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
