## [Review Process File · EMBO Molecular Medicine]

Urinary proteome profiling for stratifying patients with familial Parkinson's disease

Sebastian Virreira Winter, Ozge Karayel, Maximilian Strauss, Shalini Padmanabhan, Matthew Surface, Kalpana Merchant, Roy Alcalay, and Matthias Mann

DOI: [10.15252/emmm.202013257](https://doi.org/10.15252/emmm.202013257)

Corresponding author(s): Matthias Mann (mmann@biochem.mpg.de)

Review Timeline:

Submission Date:	8th Aug 20
Editorial Decision:	23rd Sep 20
Revision Received:	22nd Oct 20
Editorial Decision:	17th Nov 20
Revision Received:	30th Nov 20
Accepted:	10th Dec 20

Editor: Jingyi Hou

Transaction Report:

Thank you again for the submission of your manuscript to EMBO Molecular Medicine. We have now received feedback from two of the three referees whom we asked to evaluate your manuscript. Unfortunately, after a series of reminders we did not manage to obtain a report from Referee #2. In the interest of time, I prefer to make a decision now rather than further delaying the process. As you will see from the reports below, the referees acknowledge the interest and novelty of the study. However, they also raise a series of concerns about your work, which should be convincingly addressed in a major revision of the present manuscript.

Without repeating all the points raised in the reviews below, some of the most substantial issues are the following:

- Referee #1 is concerned about the FDR threshold used in the study and the small overlap in the differentially expressed protein results between the two cohorts, which need to be carefully addressed.
- Referee #1 pointed out several shortcomings of the patient cohorts. The potential limitations in this regard need to be discussed.
- Referee #3's comment on Fig 3B needs to be clarified/addressed. In particular, during our pre-decision cross-commenting process, Referee #1 added: "The two blue colors used for "bone development" and "endoplasmic reticulum lumen" in Fig 3A are quite similar on my computer screen, so it was difficult for me to confirm major point 1 by reviewer 3." In any case, we would encourage you to improve the clarity of Fig 3A, for instance, by using more visually-distinguishable colors.

All other issues raised by the referees need to be satisfactorily addressed as well. We would welcome the submission of a revised version within three months for further consideration. Please note that EMBO Molecular Medicine strongly supports a single round of revision and that, as acceptance or rejection of the manuscript will depend on another round of review, your responses should be as complete as possible.

***** Reviewer's comments *****

Referee #1 (Comments on Novelty/Model System for Author):

The model system is fine, but the number and quality of patient samples and cohort selection is suboptimal- for example, subjects are not properly age matched.

Referee #1 (Remarks for Author):

The manuscript by Winter and colleagues describes the analysis by mass spectrometry of urine samples to find biomarkers for the early diagnosis of Parkinson's disease. They used label-free, data-dependent acquisition (DIA) LC-MS/MS to characterize over 2000 proteins in 100 microliters of urine from 235 individuals in two independent cohorts of 120 and 115 subjects. Each MS run took 45 minutes, demonstrating deep urine proteome coverage using an efficient workflow. Each cohort included samples from four groups: (1) healthy controls (HC, LRRK2-/PD-); (2) non-manifesting carriers (NMC) harboring the LRRK2 G2019S mutation (LRRK2+/PD-); (3) idiopathic PD patients (iPD, LRRK2-/PD+); and (4) manifesting PD patients with LRRK2 G2019S (LRRK2 PD, LRRK2+/PD +). In the second cohort, with 115 samples from the Michael J. Fox Foundation for Parkinson's Research-funded LRRK2 Cohort Consortium, the mean age of non-diseased group was 54 years while the mean age of PD patients was 67 years, which could be problematic. The proteomic analyses were well done and robust, and although patient group size and composition were not adequate to serve in a defensible clinical trial, the manuscript demonstrates proof of principle of a potential clinically useful assay for classification of clinical PD status. The study also suggests that sample collection and handling prior to analysis may be critical, which would make establishing these assays in the clinic potentially difficult. The confirmation of the idea that potentially diagnostic CNS proteins can be found in urine is interesting.

Specific comments:

1. In both cohorts, the mean age of asymptomatic G2019S LRRK2 carriers was different from symptomatic LRRK2 carriers by 14-16 years. This age difference would make it impossible to distinguish between non-PD age-related differences and PD-related differences between symptomatic and non-symptomatic LRRK2 carriers. This seems to be a moot point however, since

few proteins changed based on PD (and therefore age) status in the LRRK2 carrier groups.

2. Several samples from each group were removed due to quality control issues, this might be difficult to replicate when testing individual samples in a clinical setting.

3. 361 proteins different between PD and non-PD (HC and NMC), 298 in Columbia cohort and 73 in LCC, implying extremely low overlap (10 proteins). This is quite troubling, and in fact demonstrates a failure of the validation cohort to verify the findings made in the discovery cohort. How can less stringent sample collection and poor age matching explain this? This implies PD status a minor driver of variation, with dire implications for eventual clinical assay development. However, the machine learning analyses demonstrate some potential utility of the assays.

4. 361 changing out of 2,000 quantified proteins is a very high percentage. Why use 5% FDR? 100 proteins would be randomly different at this FDR. Why would one expect 361/2000 urine proteins to be PD-related? How would one explain the biology behind that finding?

5. On page 14, paragraph 1 states 237 proteins changed based on LRRK2 status, while paragraph 2 states that 227 proteins changed. Please fix or explain the discrepancy.

6. Again, the overlap (33 common out of 237 total changed proteins based on LRRK2 status) seems to be quite small, suggesting that a smaller FDR than 5% should be used. Another indication is that 237/about 2000 proteins in the entire urine proteome are changed based on a single mutation- I cannot conceive of a rational biological explanation for this, especially considering the huge number of allelic variations in other proteins that must exist between the test subjects.

7. Page 17, did the TNR and FURIN levels also correlate with age? I would expect that age would also correlate negatively with cognitive performance.

8. On page 36, the "Andy West" reference is missing.

Referee #3 (Remarks for Author):

"Urinary proteome profiling for stratifying patients with familial Parkinson's disease" by Winter et al. is a novel and significant study. This study takes a powerful quantitative MS-based proteomics approach to investigate use of urine samples for molecular profiling of Parkinson's disease (PD) in two clinical cohorts. The authors have identified proteomic signatures that are associated with previously known PD mutation (LRRK2) and report potential novel urine biomarkers for PD diagnosis which could serve as a less invasive and more powerful tool compared with the existing diagnostic techniques. Additionally, the authors apply a machine learning method and show that the features selected based on the proteomic profiles of PD patients can predict PD and PD-associated mutation status of the patients. The findings are important and interesting, and the study is well done with appropriate controls. The study would certainly be of interest to the readership of EMBO Mol Med. However, to improve the manuscript there are several major and minor points that need to be addressed before publication:

Major points:

1) Fig. 3B, the color scale for "enrichment score" appears reversed. From the scatter plot in Fig. 3A, it would appear that "bone development" and "growth factor activity" are decreased in PD+ patients.

However, the "enrichment score" for these GO terms is purple or >0 and thus enriched in PD+ patients. In fact, in the discussion, the authors mention: "Most other proteins associated with the GO-term 'bone development' were downregulated in PD patients, in line with recent findings that PD patients frequently suffer from osteoporosis and osteopenia". This would again imply that the color scale has been accidentally reversed in Fig. 3A.

2) It is interesting that the GO term "Extracellular exosome" is upregulated in Fig. 3 (at least, I think it's upregulated, see note above about flipped color scale for enrichment score). Can the authors make any conclusion about whether exosomes are upregulated in PD urine? Or based on the protein identities, can the authors make any conclusion about the source of these exosomes (e.g., brain-derived, neuron-derived)?

3) In Fig. 5, the authors have calculated p-values for Pearson correlations. Have the authors applied any FDR correction to these reported p-values? If so, can the authors indicate on Fig. 6 which proteins are significant below some FDR threshold? Are these correlation coefficients available in a Supp. Table? I could not find them.

4) On Fig. 5, authors discuss TNR and FURIN, two proteins that have the most negative correlation with the MoCA score in PD+ and LRRK2+ patients. Where do TNR and FURIN lie in the proteomics analysis of PD? Are they significantly up/downregulated in the PD+ or LRRK2+ patients?

Minor points:

1) Add color scale for heatmap of Fig. 2C

2) Fig. 6: the labels are extremely small and difficult to read.

3) The numbers on the axes in Fig. 3A & Fig. 4F are difficult to read because the numbers are smooshed together.

4) Page 17: "UPDRS-III scores, which ranged from 0 to 38 in the Columbia cohort (on a scale from 0 assigned for normal to 56 for severely affected motor function)" is the range 0 to 38 or 0 to 56?

There are several typos:

Page 18: In two places, "Figure 5C" should change to "Figure 5B".

Page 18: "This protein also exhibited one the highest fold-change in abundance when comparing urine of PD patients with non-diseased individuals (Figure 3B)" should change to "This protein also exhibited one of the highest fold-changes in abundance when comparing urine of PD patients with non-diseased individuals (Figure 3A)"

Page 24: "Going forward, it would be important to examine the relationship between PD profession..." should likely be "Going forward, it would be important to examine the relationship between PD progression".

Page 25: This sentence is repeated twice: "This suggests that the genetic mutation of LRRK2 not only manifests in the central nervous systems but also dysregulates multiple pathways in distal organs such as the bladder and kidney, where LRRK2 is actually highly expressed [75]."

Point-by-point response to reviewer's comments for the manuscript 'Urinary proteome profiling for stratifying patients with familial Parkinson's disease' by Virreira Winter, Karayel et al.

Reviewer #1:

The manuscript by Winter and colleagues describes the analysis by mass spectrometry of urine samples to find biomarkers for the early diagnosis of Parkinson's disease. They used label-free, data-dependent acquisition (DIA) LC-MS/MS to characterize over 2000 proteins in 100 microliters of urine from 235 individuals in two independent cohorts of 120 and 115 subjects. Each MS run took 45 minutes, demonstrating deep urine proteome coverage using an efficient workflow. Each cohort included samples from four groups: (1) healthy controls (HC, LRRK2-/PD-); (2) non-manifesting carriers (NMC) harboring the LRRK2 G2019S mutation (LRRK2+/PD-); (3) idiopathic PD patients (iPD, LRRK2-/PD+); and (4) manifesting PD patients with LRRK2 G2019S (LRRK2 PD, LRRK2+/PD+). In the second cohort, with 115 samples from the Michael J. Fox Foundation for Parkinson's Research-funded LRRK2 Cohort Consortium, the mean age of non-diseased group was 54 years while the mean age of PD patients was 67 years, which could be problematic. The proteomic analyses were well done and robust, and although patient group size and composition were not adequate to serve in a defensible clinical trial, the manuscript demonstrates proof of principle of a potential clinically useful assay for classification of clinical PD status. The study also suggests that sample collection and handling prior to analysis may be critical, which would make establishing these assays in the clinic potentially difficult. The confirmation of the idea that potentially diagnostic CNS proteins can be found in urine is interesting.

We thank the reviewer for the positive evaluation of our manuscript and the acknowledgement of the interest of our proof-of-concept study and our findings. We are pleased to provide a detailed point-by-point response to all comments below.

Specific comments:

1. In both cohorts, the mean age of asymptomatic G2019S LRRK2 carriers was different from symptomatic LRRK2 carriers by 14-16 years. This age difference would make it impossible to distinguish between non-PD age-related differences and PD-related differences between symptomatic and non-symptomatic LRRK2 carriers. This seems to be a moot point however, since few proteins changed based on PD (and therefore age) status in the LRRK2 carrier groups.

By design of the studies, carriers of a pathogenic LRRK2 mutation without PD are typically younger than those carriers who developed the disease. Due to the low frequency of LRRK2 carriers in the general population and the increased risk of carriers to develop PD, it is challenging to include high numbers of asymptomatic LRRK2 carriers. To avoid that the age difference affects the identification of disease-related changes in pathogenic LRRK2 carriers, we performed an ANCOVA analysis with

age as one confounder variable ('bioinformatics data analysis' in the Methods). We have now included a sentence in the discussion referring to this limitation of our cohorts.

2. Several samples from each group were removed due to quality control issues, this might be difficult to replicate when testing individual samples in a clinical setting.

We thank the reviewer for this comment and confirm that some samples were excluded from the analysis. Stringent quality control is important to detect sample- or study-related biases and to avoid the reporting of false biomarkers that result from quality issues (please see our references Geyer et al. 2017 and Geyer et al. 2019). The scope of the present study was not to present a final workflow and biomarkers that can directly be applied in the clinic but rather a proof-of-concept that MS-based proteomics of urine is a valuable strategy for biomarker discovery in PD. Furthermore, it improves our understanding of PD biology and highlights pathways may be worth to be further investigated. We have now included a sentence in the discussion to discuss the need for the development of a clinical assay in the future. Building up on our data and further validation studies, specific targeted assays should be developed for clinical use.

3. 361 proteins different between PD and non-PD (HC and NMC), 298 in Columbia cohort and 73 in LCC, implying extremely low overlap (10 proteins). This is quite troubling, and in fact demonstrates a failure of the validation cohort to verify the findings made in the discovery cohort. How can less stringent sample collection and poor age matching explain this? This implies PD status a minor driver of variation, with dire implications for eventual clinical assay development. However, the machine learning analyses demonstrate some potential utility of the assays.

We thank the reviewer for highlighting the utility of our assays. We agree that the overlap is relatively small and attribute this mainly to sample heterogeneity and demographic and ethnic differences between cohorts, a common challenge in the field. Furthermore, the diagnosis of PD is mainly based on symptoms and not on biochemical tests, which could also introduce heterogeneity between the PD patients in the two cohorts and thus explain some variability in the biomarkers. Although the overlap between the two cohorts is just 10 proteins using a stringent FDR-controlled cut-off, the Pearson correlation of all 361 proteins between the two cohorts is 0.65, indicating that the majority of proteins are in fact well correlated and regulated in a similar fashion (especially in view of the above mentioned challenges). We have now added a sentence to the discussion of the revised manuscript to better explain the relatively small overlap between the two cohorts.

4. 361 changing out of 2,000 quantified proteins is a very high percentage. Why use 5% FDR? 100 proteins would be randomly different at this FDR. Why would one expect 361/2000 urine proteins to be PD-related? How would one explain the biology behind that finding?

We thank the reviewer for raising this issue. An FDR of 5% is frequently used in the bioinformatic analysis of proteomics results (as opposed to 1% FDR for peptide and protein identification) and we had fixed these statistical cut-offs before performing the analyses. Another reason not to use an FDR less than 5% is that the inter-individual variability of protein levels in urine is rather large (see Supplementary Figure 2). Based on the reviewer's question, we now repeated the ANCOVA analysis with an FDR of 1%, which resulted in 156 significantly changed PD-related proteins, which is still 5-10% of the proteome. An FDR of 5% implies that that up to 18 proteins (5% of the 361 significantly regulated proteins) are potentially false-positive hits (rather than 100 seemingly but not actually changing proteins of the total of 2000).

We agree that the number of PD-regulated proteins is rather large but can only speculate at this point about the underlying biology. The fact that up to 85% of PD patients have urinary tract dysfunction (see Yeo et al. Int Urol Nephrol, 2012) could well explain the observed proteome changes.

5. On page 14, paragraph 1 states 237 proteins changed based on LRRK2 status, while paragraph 2 states that 227 proteins changed. Please fix or explain the discrepancy.

We thank the reviewer for spotting this mistake. The ANCOVA analysis revealed 237 significantly changed proteins based on LRRK2 status as also shown in Figure 4A. We have fixed this typo in the revised manuscript.

6. Again, the overlap (33 common out of 237 total changed proteins based on LRRK2 status) seems to be quite small, suggesting that a smaller FDR than 5% should be used. Another indication is that 237/about 2000 proteins in the entire urine proteome are changed based on a single mutation- I cannot conceive of a rational biological explanation for this, especially considering the huge number of allelic variations in other proteins that must exist between the test subjects.

This question relates to the one just above – please see explanation of the 5% cut-off. When we repeated the ANCOVA analysis with an FDR of 1%, we obtained 111 LRRK2-regulated proteins with an overlap of 20 between the two cohorts, strongly supporting that the mutation causes large changes in the proteome.

In our view, the observation that a single point mutation like the LRRK2 G2019S mutation analyzed in the present study result in significant abundance changes for a large number of proteins is not necessarily unexpected or unusual. In support of this, a recent study (Connor-Robson et al. 2019) using an integrated transcriptomics and proteomics approach found that more than 2,000 proteins and 2,000 genes were differentially expressed in iPSCs-derived dopaminergic cultures harboring the LRRK2 G2019S mutation compared to wild type LRRK2 cultures. Apparently, the significant dysregulation of membrane trafficking, exosome biology and lysosomal function caused by the pathogenic LRRK2 mutation in turn causes large changes in the proteome.

7. Page 17, did the TNR and FURIN levels also correlate with age? I would expect that age would also correlate negatively with cognitive performance.

TNR and FURIN levels are actually not correlated with the age at sample collection. The Pearson correlations are 0.19 (TNR) and 0.16 (FURIN) among all PD+ individuals and 0.20 (TNR) and 0.24 (FURIN) among all LRRK2+/PD+ individuals. We have now included this information also in the revised results section. However, as the reviewer expected, there is a weak negative correlation of age with cognitive performance. The Pearson correlation of age at sample collection vs. MoCA total score is -0.41 in PD+ individuals (see figure below) and -0.54 in LRRK2+/PD+ individuals.

8. On page 36, the "Andy West" reference is missing.

We thank the reviewer for pointing out the missing reference and the revised version now includes a citation from the laboratory of Andy West there. We now also included other citations from the same laboratory in the main text.

Reviewer #3:

Urinary proteome profiling for stratifying patients with familial Parkinson's disease" by Winter et al. is a novel and significant study. This study takes a powerful quantitative MS-based proteomics approach to investigate use of urine samples for molecular profiling of Parkinson's disease (PD) in two clinical cohorts. The authors have identified proteomic signatures that are associated with previously known PD mutation (LRRK2) and report potential novel urine biomarkers for PD diagnosis which could serve as a less invasive and more powerful tool compared with the existing diagnostic techniques. Additionally, the authors apply a machine learning method and show that the features selected based on the proteomic profiles of PD patients can predict PD and PD-associated mutation status of the patients. The findings are important and interesting, and the study is well done with appropriate controls. The study would certainly be of interest to the readership of EMBO Mol Med. However, to improve the manuscript there are several major and minor points that need to be addressed before publication:

We thank the reviewer for the positive and supportive evaluation of our work - acknowledging the potential of MS-based urinary proteomics for biomarker discovery and patient stratification. We also thank the reviewer for the useful comments and are pleased to provide the point-by-point response to each comment below.

1. Fig. 3B, the color scale for "enrichment score" appears reversed. From the scatter plot in Fig. 3A, it would appear that "bone development" and "growth factor activity" are decreased in PD+ patients. However, the "enrichment score" for these GO terms is purple or >0 and thus enriched in PD+ patients. In fact, in the discussion, the authors mention: "Most other proteins associated with the GO-term 'bone development' were downregulated in PD patients, in line with recent findings that PD patients frequently suffer from osteoporosis and osteopenia". This would again imply that the color scale has been accidentally reversed in Fig. 3A.

We thank the reviewer for highlighting the unclear explanation of the GO-term enrichment analysis. We rephrased the paragraph in order to avoid any confusion.

Fig. 3A shows the direction of changes for all PD-regulated proteins, i.e. all proteins that have significantly different levels in urine of PD patients compared non PD individuals. The GO-term enrichment analysis determines if any GO-terms associated with these 361 PD-regulated proteins are significantly enriched or de-enriched compared to the total urinary proteome of about 2,000 proteins. The Fisher exact test does not take the direction of change into account. Indeed, while the GO-terms 'bone development' and 'growth factor activity' are enriched in PD-regulated proteins, many proteins associated with these terms are decreased in PD patients.

2. It is interesting that the GO term "Extracellular exosome" is upregulated in Fig. 3 (at least, I think it's upregulated, see note above about flipped color scale for enrichment score). Can the authors make any conclusion about whether exosomes are upregulated in PD urine? Or based on the protein identities, can the authors make any conclusion about the source of these exosomes (e.g., brain-derived, neuron-derived)?

As stated above, the enrichment analysis does not allow to draw any conclusions about the direction of regulation. Our data do not indicate that exosomes are up- or downregulated in the urine of PD patients. The fact that the GO-term 'extracellular exosome' is actually de-enriched among the 361 PD regulated proteins indicates that they are actually not associated with urinary exosomes. However, a

more detailed investigation of exosome synthesis and secretion may be useful in future studies. We have now included a sentence in the results section referring to this point.

3. In Fig. 5, the authors have calculated p-values for Pearson correlations. Have the authors applied any FDR correction to these reported p-values? If so, can the authors indicate on Fig. 6 which proteins are significant below some FDR threshold? Are these correlation coefficients available in a Supp. Table? I could not find them.

We thank the reviewer for this comment. The p-values shown in Figure 5 are before correction for multiple testing. However, we have now highlighted all proteins that were significantly correlated when we used a 5% FDR cutoff using the Benjamini-Hochberg correction. The correlation scores including p- and q-values can be found in a newly added Supplementary Table 6.

4. On Fig. 5, authors discuss TNR and FURIN, two proteins that have the most negative correlation with the MoCA score in PD+ and LRRK2+ patients. Where do TNR and FURIN lie in the proteomics analysis of PD? Are they significantly up/downregulated in the PD+ or LRRK2+ patients?

We thank the reviewer for this comment. Neither TNR nor FURIN were significantly regulated in the urine of PD patients (TNR q-value: 0.31/0.65 in Columbia/LCC; FURIN q-value: 0.08/0.66 in Columbia/LCC) or pathogenic LRRK2 carriers (TNR q-value: 0.90/0.93 in Columbia/LCC; FURIN q-value: 0.10/0.83 in Columbia/LCC). We have now included a sentence in our discussion to clarify this observation.

Minor:

1. Add color scale for heatmap of Fig. 2C

We thank the reviewer for noticing the missing color scale, which we now included in the revised Figure 2.

2. Fig. 6: the labels are extremely small and difficult to read.

We have now increased the font sizes in Figure 6.

3. The numbers on the axes in Fig. 3A & Fig. 4F are difficult to read because the numbers are smooshed together.

We have now clearly separated the numbers from each other in Figure 3A and 4F.

4. Page 17: "UPDRS-III scores, which ranged from 0 to 38 in the Columbia cohort (on a scale from 0 assigned for normal to 56 for severely affected motor function)" is the range 0 to 38 or 0 to 56?

We thank the reviewer for highlighting the unclear phrasing in our manuscript. We have rephrased the paragraph and updated the numbers for the range in order to clarify the UPDRS-III scoring in the revised version.

5. Page 18: In two places, "Figure 5C" should change to "Figure 5B".

We have corrected the figure callouts for Figure 5B and 5C in the revised manuscript.

6. Page 18: "This protein also exhibited one the highest fold-change in abundance when comparing urine of PD patients with non-diseased individuals (Figure 3B)" should change to "This protein also exhibited one of the highest fold-changes in abundance when comparing urine of PD patients with non-diseased individuals (Figure 3A)"

We have changed the sentence in the revised manuscript.

7. Page 24: "Going forward, it would be important to examine the relationship between PD profession..." should likely be "Going forward, it would be important to examine the relationship between PD progression".

We have corrected the sentence in the revised manuscript.

8. Page 25: This sentence is repeated twice: "This suggests that the genetic mutation of LRRK2 not only manifests in the central nervous systems but also dysregulates multiple pathways in distal organs such as the bladder and kidney, where LRRK2 is actually highly expressed [75]."

We have deleted the misplaced sentence in the revised manuscript.

Thank you for the submission of your revised manuscript to EMBO Molecular Medicine. We have now received the enclosed report from two referees who were asked to re-assess it. As you will see the referees are now overall supportive and I am pleased to inform you that we will be able to accept your manuscript pending the following amendments: 1. Both referees still raise a couple of concerns regarding the statistics and data analysis, which need to be addressed and clarified.

***** Reviewer's comments *****

Referee #1 (Comments on Novelty/Model System for Author):

The technical quality of the mass spectrometry analysis is high but the composition of the study cohorts has limitations which are mentioned in the text. The paper is useful as a demonstration of potential utility of studying urinary proteins by mass spectrometry, but is not adequate to serve as an actual clinical study. Human patients are used rather than a model system.

Referee #1 (Remarks for Author):

The authors have made some changes in response to critiques and manuscript has been improved as a result. However, it would be helpful to include a few sentences to the text in case some readers have the same questions as this reviewer. Specifically:

1. In response to Specific Comment 4, please mention that up to 85% of PD patients have urinary tract dysfunction and cite the Yeo reference in the section where the result that 361/2000 urine proteins changed in PD patients. Other readers may wonder why the number of changed proteins is so high (a more worrying explanation would be poor experimental execution and/or data analysis).
2. In response to Specific Comment 6, a Pearson correlation should be calculated and reported for the two LRRK2 cohorts because overlap (33/237 at 5% FDR) of changed proteins between the cohorts is so low- this is a similar problem as with the PD cohorts mentioned in Comment 4 and should be addressed in a similar manner. Again, a sentence explaining why the LRRK2 mutation could cause so many protein changes should be added to the text along with the Connor-Robson reference.

Referee #3 (Remarks for Author):

The authors have sufficiently addressed my initial concerns.

I still find it quite confusing that the Fisher's Exact Test in Fig. 3 "does not take the direction of change into account." The presence of the log₂ Enrichment Score (in my mind) implies a directionality of enrichment. However, if the authors could add a sentence or two (perhaps in the figure legend) being a bit more explicit that the enrichment score refers to PD-regulated proteins v. non-PD-regulated proteins and therefore the enrichment score does not reflect which proteins are up/down in PD, that would be helpful.

Point-by-point answers to ‘Urinary proteome profiling for stratifying patients with familial Parkinson’s disease’ by Virreira Winter and Karayel et al.

Reviewer's comments

Referee #1 (Comments on Novelty/Model System for Author):

The technical quality of the mass spectrometry analysis is high but the composition of the study cohorts has limitations which are mentioned in the text. The paper is useful as a demonstration of potential utility of studying urinary proteins by mass spectrometry, but is not adequate to serve as an actual clinical study. Human patients are used rather than a model system.

We thank the reviewer for these kind words and thank for the useful comments that helped us to strengthen our manuscript.

Referee #1 (Remarks for Author):

The authors have made some changes in response to critiques and manuscript has been improved as a result. However, it would be helpful to include a few sentences to the text in case some readers have the same questions as this reviewer. Specifically:

1. In response to Specific Comment 4, please mention that up to 85% of PD patients have urinary tract dysfunction and cite the Yeo reference in the section where the result that 361/2000 urine proteins changed in PD patients. Other readers may wonder why the number of changed proteins is so high (a more worrying explanation would be poor experimental execution and/or data analysis).

In response to the reviewer’s comment, we have now included a sentence in the section stating that the majority of PD patients show urinary tract dysfunction and refer to the previous study of Yeo et al.

“Applying a 5% false discovery rate (FDR) cut off, we identified 361 proteins that displayed significantly different levels in PD patients when compared to controls (HC and NMC) (298 in Columbia cohort and 73 in LCC cohort) (**Dataset EV5**). **The relatively large number of regulated proteins is in agreement with previous reports that the majority of PD patients suffers from urinary tract dysfunction (Yeo, Singh et al., 2012).** The smaller number of significantly different proteins in

the LCC cohort as well as the relatively small overlap between the cohorts could be explained by a less stringent sample collection protocol and worse age-matching in the LCC cohort.”

2. In response to Specific Comment 6, a Pearson correlation should be calculated and reported for the two LRRK2 cohorts because overlap (33/237 at 5% FDR) of changed proteins between the cohorts is so low- this is a similar problem as with the PD cohorts mentioned in Comment 4 and should be addressed in a similar manner. Again, a sentence explaining why the LRRK2 mutation could cause so many protein changes should be added to the text along with the Connor-Robson reference.

In response to the reviewer’s comment, we have now included sentences in the results section to report the Pearson correlation between the two studies. We also now included the Connor-Robson reference in the discussion, to better explain why the LRRK2 mutation may cause the observed proteome changes.

“In total, 237 LRRK2 status-associated proteins were quantified in both cohorts and the fold-changes of these were similar between the two cohorts (Pearson correlation: 0.34, Figure 4C). The lower correlation between the studies compared to the PD-associated proteins could be explained by the different distribution of PD+ and PD- individuals in the two cohorts, which is only corrected for in the applied ANCOVA analysis but not in this Pearson correlation analysis. The effect sizes of the LRRK2-associated proteins were slightly larger for the Columbia cohort (Columbia: 1.43 (up) & 0.76 (down) vs. LCC: 1.39 (up) & 0.89 (down)).”

„Our results demonstrate that urine of pathogenic LRRK2 carriers strongly reflects lysosomal dysregulation associated with increase in LRRK2 activity (Alessi & Sammler, 2018). These major proteome changes are in agreement with a previous study that observed more than 2,000 proteins to be affected by pathogenic LRRK2 (Connor-Robson, Booth et al., 2019). One of the strongest upregulated proteins in LRRK2 G2019S carriers was the alkaline phosphatase ALPI.“

Referee #3 (Remarks for Author):

The authors have sufficiently addressed my initial concerns.

I still find it quite confusing that the Fisher's Exact Test in Fig. 3 "does not take the direction of change into account." The presence of the log2 Enrichment Score (in my mind) implies a directionality of enrichment. However, if the authors could add a sentence or two (perhaps in the figure legend) being a bit more explicit that the enrichment score refers to PD-regulated proteins v. non-PD-regulated proteins and therefore the enrichment score does not reflect which proteins are up/down in PD, that would be helpful.

We thank the reviewer for helping us to improve our manuscript and appreciate that the initial concerns were sufficiently addressed. In response to the reviewer’s comment, we have now included a sentence in the figure legend of Figure 3 to clarify that the enrichment score doesn’t indicate any direction of the regulation.

“Importantly, the enrichment score of the Fisher exact test does not indicate if the proteins were up- or downregulated in PD patients but rather that the regulated proteins – independent of the directionality – compared to the total urinary proteome are associated with the enriched term.”

Accepted

10th Dec 2020

We are pleased to inform you that your manuscript is accepted for publication and is now being sent to our publisher to be included in the next available issue of EMBO Molecular Medicine.

Corresponding Author Name: Matthias Mann

Manuscript Number: EMM-2020-13257